# Small Houses, Big Community: Tiny Housers' Desire for More Cohesive and Collaborative Communities

**Chelsey Willoughby [1], Severin Mangold [2]**  **and Toralf Zschau [1,*]**

1    Department of Sociology & Human Services, University of North Georgia, Dahlonega, GA 30597, USA; clwill7486@ung.edu
2    Department of Sociology, Colorado State University, Fort Collins, CO 80523, USA; smangold@colostate.edu
*    Correspondence: tzschau@ung.edu

**Abstract:** Past research on the tiny house movement has primarily focused on understanding the individual motivations behind adopting the tiny house lifestyle. While some studies have suggested that tiny housers do entertain an interest in community, no systematic research exists that examines the actual complexities of this phenomenon. To make first inroads into this body of literature, twenty-four community-oriented tiny housers were interviewed about their ideal community. Interview questions ranged from definitions of community to specific ideas of the nature of community characteristics. Interviews were recorded, transcribed, and then coded in NVivo 12.0. Four main themes and eleven subthemes emerged from the qualitative content analysis. Select themes were then subjected to a subsequent quantification analysis in order to refine and deepen the theoretical understanding. The findings of this exploratory study suggest that a majority of tiny housers desire to be part of more cohesive and collaborative communities. While stressing the importance of community, tiny housers also expressed concerns over privacy. To explain the findings, the paper offers a set of arguments situated in the broader socio-cultural texture of our time.

**Keywords:** tiny house; alternative lifestyle; community; simple living; privacy

---

## 1. Introduction

Over the past two decades, the contemporary Tiny House[1] (TH) movement has gained traction around the world and is now increasingly seen as a transnational phenomenon (BBCNews 2019; Browne 2019; McNamee 2019). The early contours of this lifestyle movement[2] emerged within the United States in the late 1980s and early 1990s[3] as a quasi-countercultural response to American hyper-consumerism, as well as an attempt to find more tangible solutions for affordable housing (Carlin 2014). In the early 2000s, many of these ideas crossed the Atlantic to Europe (e.g., Germany, Sweden, and the Netherlands), where individuals interested in sustainable living began to rediscover the benefits of small spaces and micro-living (Schneider 2017). It was not until the 2007–2008 housing

---

1    While disagreements over an exact definition continue to exist, the term tiny house usually refers to living spaces smaller than 400sqft. Tiny housers have built tiny houses on wheels, on a foundation and/or have explored ways to repurpose old school buses, houseboats or vans. However, the literature increasingly distinguishes between tiny houses and other forms of small living arrangements (e.g., micro apartments, micro units). For further information see (Shearer and Burton 2018).
2    The authors of this manuscript follow the theoretical distinction made between lifestyle movements and social movements (Haenfler et al. 2012) by treating the TH movement as a lifestyle movement.
3    The contemporary tiny house movement is often traced to the writings of Lloyd Kahn's 1973 "Shelter", Les Walker's 1987 "Tiny, Tiny Houses" or Sara Susanka's 1997 "The Not so Big House," and the advocacy of movement pioneers like Jay Shafer—though the exact lineage remains difficult to pin down (for a more detailed treatment of the subject please see April Anson (2017, 2018) brilliant work.

crisis and global financial meltdown, however, that the US-based TH movement inscribed itself onto the larger psyche of popular culture. Widespread mortgage foreclosures and rising unemployment rates made many people not only question the American dream (Anson 2018; Carlin 2014), but also pushed them to seek lifestyle alternatives.

The US-based tiny house movement appropriates ideas that can be traced to Thoreau's transcendentalism, voluntary simplicity, and minimalism (Anson 2017, 2018; Ford and Gomez-Lanier 2017; Harris 2018). In its commodified pop-culture-esque version, the movement echoes aspects of the coming-of-age-story of the United States with its frontier ethos, rugged individualism, white settler colonialism, and to some extent elite environmentalism (Anson 2018). By portraying an existentialist pilgrimage "down an empty freeway" in "search of romantic landscapes," "freedom," and "unoccupied land," the more visible THOW[4] tradition within the TH movement in particular resurrects America's settler mythology and reaffirms the value of private property (Anson 2018). Some scholars argue that the TH house lifestyle constitutes a "privileged" movement because more advantaged individuals are able to voluntarily make the choice to downsize (Anson 2014; Schneider 2017). Others contend that people often transition into the lifestyle because they feel *they have to*. Drawing on series of articles, they maintain that individuals frequently report financial struggles before they decided to adopt the lifestyle (Hanckmann 2019; Harris 2018). Conceptual debates like these highlight that the movement is not monolithic, but that it captures a wide a range of complex lifestyle choices (Harris 2018). While some scholars view the environmental (Ford and Gomez-Lanier 2017; Kilman 2016; Vannini and Taggart 2016) and anti-consumerist aspects (Harris 2018; Roy 2019) as the ideological core of the movement, others argue that the TH lifestyle, at its heart, constitutes an existential quest for the "Good Life." Mangold and Zschau (2019), for example, believe that many tiny housers simply embrace this lifestyle to regain a sense of ontological security and individual autonomy in order to define life on their terms. Whether individuals embrace the lifestyle out of necessity or choice, all try to move past the cultural obsession with "stuff" in order to embark on a journey in search of new experiences and/or more meaningful relationships (Hanks 2017). They also prescribe to a "less is more" philosophy (Ford and Gomez-Lanier 2017; Mangold and Zschau 2019)—a philosophy that has increasingly found converts in other (especially "first world") countries.[5]

While the mainstream media often depicts tiny housers as rugged individuals with semi-nomadic lifestyles, the larger movement contains a significant segment of more community-oriented tiny housers. Boeckermann (2017) for example, found that for a majority of tiny housers, having a sense of community is an "important" factor for *living tiny* (with a quarter of them seeing it as a "highly" important). Other studies echo this desire for social connectedness by suggesting that tiny housers seek meaningful relationships and stress the importance of community in their lives (Harris 2018; Mangold and Zschau 2019). Kilman (2016) also stresses that living tiny not only creates new opportunities for people to interact and/or get involved within their local communities, but that the lifestyle can also foster more intimate encounters with nature. This desire for community can also be seen, at least implicitly, in the creation of TH communities in states like California, Texas, and Florida (Sullivan 2019). Today, there are a total of 98 registered TH communities in the United States

---

[4]  It is estimated that there are over 10,000 tiny houses in the United States, with more being built each year (IPropertyManagement 2019). It is less clear, however, to what extent this number includes Tiny Houses on Wheels (THOWs). The actual number of tiny houses may thus be substantially larger.

[5]  Two recent studies from Germany, for example, illustrate that German tiny housers also share the "American" emphasis on autonomy, improved relationships, and 'self-reliance' (Böllert 2019; Hanckmann 2019). Penfold et al. (2018), in turn, have illustrated how Australian tiny housers have begun to challenge the longstanding obsession with McMansions, and instead opt for a similar 'less is more' philosophy and smaller places to live. Dutch tiny housers, like some of their American, German and Australian counterparts, also show a strong interest in simple and sustainable living (Böllert 2019). Many became interested in the tiny house lifestyle in the aftermath of the global financial crisis of 2007–2008 and a more general struggle with affordable housing (Schneider 2017). Other sources have documented the emergence of a homegrown tiny house movement in countries such as Finland (Browne 2019), Switzerland (Bondolfi 2019), New Zealand (Wilkes 2019), and The UK (BBCNews 2019; McNamee 2019).

(Tiny Home Industry Association 2019), and an increasing number of municipalities are considering changing land use laws and building codes to allow for tiny houses (Koch 2015). Tiny housers, like individuals in other lifestyle movements (Huneke 2005), have also formed loose but active online communities. They use blogs, online forums, and websites to share information, which may help reinforce TH identities (Hutchinson 2016; Katra 2017). Despite the growing popularity of the tiny house lifestyle (Ro 2018), however, there is little or no systematic research on the types of community tiny housers would actually want to live in. Much of what is known remains conceptually underdeveloped, anecdotal, or outright speculative.

To offer new conceptual inroads, the following paper documents insights from a US-based exploratory study that examines two interrelated research questions: (1) *What are tiny housers' views of community*? and (2) *What specific community characteristics and dynamics are important to them?* To better contextualize the findings, the paper begins with a short literature review that (1) discusses what community is and how community has changed, and then (2) summarizes views on community in related lifestyle movements and intentional communities. After having sketched out the basic components of the research design, the paper will discuss insights from 24 semi-structured interviews with community-oriented tiny housers. The findings suggest that while all tiny housers want to live in communities with similar social and spatial/infrastructural characteristics, they differ on how deeply they want to engage with others. Subscribing to one of three community engagement styles, the tiny housers in this study articulated a particular view of community (mainstream, cohesive, or collaborative), with a majority being drawn to more cohesive and collaborative communities. These place-based communities are characterized by a much stronger focus on meaningful social relations, reciprocal social support systems, active communal spaces, high-demand community activities, and an expectation that resources and skills will be shared among members. At the same time, however, tiny housers also expressed privacy concerns and wanted to know that their spatial and behavioral autonomy would be safeguarded. Situating the results within the larger socio-cultural matrix of our time, the paper will argue that these community preferences are driven by a perceived loss of community, and/or an increasing exposure to alternative forms of community. The documented concerns over privacy, it will be argued, may not only be an outcome of the psycho-spatial realities of tiny living, but also a reflection of the social construction of privacy in the West. The paper concludes by offering a number of broad suggestions for future research, policy, as well as community development.

## 2. Literature Review

### 2.1. What is Community?

Definitions of community vary widely across the literature, and an agreement over its precise meaning remains elusive (Goe and Noonan 2007). Scholars continue to debate if geographic proximity is a necessary condition for community or whether communities can transcend "time" and "place". Theoretical differences also exist over the *appropriate geographic scale* (e.g., the neighborhood, a small town, a city, a nation, or the global community), the *precise nature of social relations* (e.g., strong ties versus weak ties), and the *extent to which communities are able to construct social identities* for its members (Goe and Noonan 2007; Gusfield 1975; Hillery 1955; Piselli 2007; Poplin 1979).

These conceptual difficulties have led to the emergence of three major theoretical traditions which emphasize that community has either been *lost* (Durkheim 1997; Etzioni 1999; Putnam 2000; Tönnies 1988) [6], *saved* (Wellman 1979), or *liberated* (Beck 1994, 2000; Wellman and Haythornthwaite

---

[6] Tönnies ([1887] 1988) (and later Durkheim, 1893), for example, argued that 19th century industrialization and urbanization radically altered the nature of European "communities". Smaller, more rural forms characterized by strong emotional attachments, intimacy, and shared kinship that were held together by strong religious values and beliefs (called Gemeinschaft) began to give way to larger, instrumental, impersonal and delocalized forms of human settlements (termed Gesellschaft). This transformation not only changed the nature of social cohesion within communities, but also fundamentally changed

2002). *Community Lost* perspectives stress that social and political changes often translate into a loss of place-based communities and an erosion of strong reciprocal ties, which in turn, lead to more impersonal, transitory, and highly fragmented social relations (especially in urban and suburban contexts). The *Community Saved* tradition, in contrast, argues that community has not been lost, but rather has shifted from kinship-based networks to more residential and work-related networks. These emerging networks create new forms of "communal solidarities" that continue to offer "support and sociability, [and] communal desires for informal social control". The final perspective, the *Community Liberated* framework (and its conceptual offshoots) contends that the meaning of community was "liberated" from place and now takes on the form of multiple "sparsely knit, spatially dispersed," and only loosely connected sets of friendship and family networks (Wellman 1979). These new communities are also seen as "liberated" or "liberating" because they allow individuals to overcome the oppressive and stifling aspects of traditional place-based communities (Cohen 1985; Sprigings and Allen 2005).

Rather than being forced to live in a community that was, in a sense, imposed on the individual, these newer forms of community allow a person to choose what type of community they want to live in. Scholars have argued that we are not only witnessing a "globalization of biography" (Beck 2000) and an emergence of "elective biographies," but also a fundamental transformation of the meaning of place in peoples' lives (Jamieson 1998; Urry 2000). Today, "friends" and "neighbors" constitute social networks spread across the globe stringing together individuals that often live multiple hemispheres and time zones away (Beck 1994, 2000). While this may seem to promise communities of choice (Beck 1994), there is a real danger that the erosion of local place-based communities will exacerbate social ills and lead to more anonymity, individualism, and feelings of social isolation, especially in urban or suburban areas (Forrest and Kearns 2001). Bradshaw (2008) appropriately calls these emerging networks of global solidarity, shared identities, and social norms "post-place communities." Whether community is treated as a conceptual ideal-type or a lived reality for social actors, meanings differ—often widely—across time, space, and even subpopulations. In a sense, views on community are not static but are constantly being renegotiated within the context of larger social and cultural forces (Giddens 1990; Goe and Noonan 2007).

### 2.2. Community in Other Lifestyle Movements or Intentional Communities

The tiny house movement has drawn much inspiration from other lifestyle movements as well as intentional communities (Mangold and Zschau 2019). While there is a lack of systematic research on how community is viewed within these sister movements, the following section attempts to paint a simple picture of the role community plays in *Voluntary Simplicity* (VS), *Minimalism*, and certain *intentional communities* (e.g., Cohousing Communities and Ecovillages).

### 2.2.1. Voluntary Simplicity and Minimalism

Herrier and Murray (2002) argue that minimalism and voluntary simplicity are distinct movements that may share some of the same grievances but offer two very different anti-consumerist pathways. Both stress that removing "excess" from individuals' lives and making lifestyle changes are crucial milestones on the road to a good life. They differ, however, on the role they impart on environmental sustainability. While being a core theme for voluntary simplicity, saving the planet and reducing one's ecological footprint are only secondary concerns in minimalism (Meissner 2019). Despite these more "individualistic" life prescriptions, many adhering to these lifestyle movements entertain, however weak, views of community.

Voluntary simplicity (VS) as a lifestyle movement, encourages "a manner of living that is outwardly more simple and inwardly more rich" (Elgin 1977). While VS, like minimalism, centers on issues of

---

how individuals related to one another. What is interesting here is that many of the core issues raised by these classic sociologists continue to animate today's academic debates.

individual agency, personal wellbeing, and fostering deeper relationships with family members and friends (Kraisornsuthasinee and Swierczek 2018), community does matter. Many voluntary simplifiers are part of VS circles and discussion forums, most of which are "web-based" (Huneke 2005). VS circles and other online forums provide social spaces for newcomers and long-time VS practitioners to interact, share ideas, as well as offer support to one another (Ibid.). Grigsby (2004), in contrast, contends that VS—at its core—consists of a daily re-commitment to building "local networks of relationships that will endure over time." The emphasis on social ties also means that contemporary voluntary simplifiers are often found to be actively involved with their wider communities. Studies have shown, for example, that 67% of voluntary simplifiers work with a community organization, and 41% with a community or political organization "related to simple living", either locally, regionally, or nation-wide (Alexander and Ussher 2012). In short, embracing this particular countercultural lifestyle often means that individuals straddle both worlds, that of traditional place-based and those of more modern post-place communities. The loose connections among participants within the movement and the nature of their community work suggests, however, that voluntary simplifiers tend to live in more *liberated* forms of community. To what extent this represents an "ideal" community for these individuals, though, is less clear since much of the literature focuses on existing community ties.

Minimalism, another lifestyle movement that shares common ideological themes with the TH movement, has gained much popularity recently (Babauta 2009; Millburn and Nicodemus 2015). The authors frame minimalism as a life philosophy that provides individuals the tools to remove "unnecessary things" from their lives and enables them to "focus on what's important" (Millburn and Nicodemus 2015). By shifting from "stuff" to "experiences" and adhering to a strict "less is more" ideology (Wideman 2019), the lifestyle promises its adherents a path to true happiness (Amanda 2017; Becker 2015; Mangold and Zschau 2019), a roadmap for pursuing one's passions (Hanks 2017), and a chance to live a better life in the here and now (Altucher 2016). Today, many websites, blogs, or books promote the lifestyle with self-help-esque advice, minimalist workshops, and informational videos (Feedspot 2019).

Despite its growing popularity, there is little systematic academic research on the role community plays in the lives of minimalists. A short sample of existing minimalist material, however, suggests that web-based spaces such as www.theminimalists.com, www.becomingminimalist.com, reddit.com/minimalism, and social-media-driven "Meetup" groups have a central place. Minimalists (and those who are interested in the lifestyle) meet in these groups to discuss best practices or share resources. While some minimalists do promote prosocial concerns in their communities or in the world (Wideman 2019), collective concerns tend to be de-emphasized. Minimalists, like voluntary simplifiers, interact with one another in online spaces such as Facebook groups (e.g., *Our Minimalist Lifestyle, Practical Minimalism,* or *Slow, Eco, Simple Minimal Living*). However, the dis-embedded nature of these post-place virtual communities and the more individualist core of minimalist philosophy probably lead to more fluid, and possibly, more instrumental views of community: communities as a utilitarian means toward individual happiness. Future research will need to disentangle these difficult complexities.

### 2.2.2. Intentional Communities

Contemporary intentional communities such as cohousing initiatives, ecovillages, and urban housing cooperatives constitute efforts to mitigate the perceived loss of community and quality of life by reestablishing *Gemeinshaft*-esque community relations or by creating functional homologies to close-knit families. Intentional communities involve people that make an "intentional" decision to live together. By embracing a common purpose, working cooperatively, and/or living a lifestyle that reflects members' shared values and beliefs (Smith 2002), people in these communities not only reaffirm the importance of place for nurturing healthy social relations, but also the need for more collaborative communal activities.

Cohousing communities, for example, stress shared facilities such as common houses or common spaces (Williams 2005). They are built to facilitate vibrant social interactions among members, adhere

to the principle of self-administration, and have mission statements that often stress environmental stewardship (Daly 2017). Homes are built at higher densities to save space for shared facilities (e.g., community kitchens, laundromats) or other common spaces that can nurture human connection among its members (Assadourian 2008; Ruiu 2014). The unique focus on social relationships, collaboration, and shared resources in the age of hyper-consumerism and neighborhood erosion is what makes the cohousing model, for some, a blueprint for the *ideal* community (Sargisson 2012). For ecovillages, on the other hand, environmental sustainability is the central focus (Smith 2002). Ruiu (2014) places ecovillages within the global environmental movement, because for her they constitute a systemic effort to encourage simpler lifestyles, promote environmentally sustainable practices, and design communities that work with nature, rather than against it. Ecovillages employ ecofriendly physical and architectural designs, foster meaningful social interactions among its members, and adhere to the democratic principles of self-governance. Many ecovillagers also engage in community outreach efforts (Van Schyndel Kasper 2008) and/or are active in environmental organizations (Gray 2008). While ecovillagers seem to build communities informed and guided by strong ecological principles, co-housers put a higher premium on community cohesion and member involvement (which may or may not involve a systematic commitment to sustainability).

In short, it seems that the individuals who live in intentional communities value community cohesion, social relationships, collaboration, and community involvement more strongly than those in other lifestyle movements. To them, community is not merely a geographic place or an abstract idea, but an existentialist fulcrum around which *all* meaningful life revolves.

## 3. Methods

To understand tiny housers'[7] views of community, a quasi-stratified purposive sample of twenty-four tiny housers in the United States (US) interested in living in "some form" of community was created (Creswell 1998; Patton 2002). Participants were recruited through an email list from a regional tiny house builder and several TH groups on Facebook and Reddit (i.e., *Tiny House People, Tiny House Life, DFW Tiny House Enthusiasts).* This approach created a pool of potential participants. To not only capture differences in views, but to also create an adequate sample[8], the researchers created a "gender strata" (twelve males and twelve females) and an "age strata" (with eight individuals falling into one of three age categories (18–34, 35–54, 55+). As a result, the sample captures a cross-section of the overall US tiny house movement. It includes tiny housers from eleven different states and individuals who differ in terms of their income, political orientation, educational levels, geographic location (urban, suburban, rural), and religious orientation (see Appendix A for more details).

---

[7]  A tiny houser is defined here as an individual who (1) has made a concerted effort to learn about the tiny house lifestyle, (2) currently lives in a tiny house, or (3) is in the planning, building, or buying stage of a tiny house.

[8]  Given the absence of an adequate sampling frame, non-probability sampling techniques were used for this research. To improve the overall representativeness of the sample, however, quasi-stratified purposive sampling was chosen over simple convenience sampling. This technique is generally used to "capture major variations [within the population] rather than to identify a common core, although the latter may also emerge in the analysis." (Patton 2002, p. 240). The final size of the sample for this research was informed by three major considerations: (1) Creswell (1998) recommendations for interview research (i.e., 20–30 participants), (2) the purposive sampling rationale, and (3) the conclusion that thematic and data saturation may have been reached during the coding process, suggesting no further need for additional interviews (Francis et al. 2010; Saunders et al. 2018). We acknowledge that this approach may not have yielded a perfectly representative sample, and that there might be a hidden selection bias (e.g., Some participants in the sample were recruited from social media and thus may present a more community-oriented subpopulation). However, the social profile of our participants and the nature of prior research, which suggests that the TH community is primarily located online (Katra 2017), makes us believe that the sample still constitutes a "fair" approximation. While little is known about the overall demographic makeup of movement, the fact that the social characteristics of our participants compare to those suggested by others lends at least some credence to this claim. The existing research generally suggests that the "average" tiny houser in the United States tends to be white, more educated and able to command household incomes that cuts across the income spectrum. (e.g., Boeckermann 2017, MicroLife Institute Atlanta Personal Communication, Authors: Unpublished Data, Harris 2018). Given these methodological complications, however, we invite the reader to draw their own conclusions as to the generalizability of the findings.

Semi-structured phone interviews were conducted during the summers of 2017 and 2019. Each participant was asked for consent prior to being interviewed and pseudonyms were assigned to protect identities. The interviews covered a range of topics but primarily explored tiny housers' views of community. Interviewees were prompted with broad questions such as: *What does community mean to you? What are you looking for in a community*? Later sections of the interviews dove into questions about more specific community aspects such as *What is the preferred size and location for your community? What kind of governance structures would you like to see in your community? What, if any, communal spaces, community activities, and shared resources would you like to have in the community? And Why?* Probes were directed toward unveiling concerns for privacy, and the level of engagement being sought within community. All interviews (average length: 73 min) were recorded, transcribed in NVivo 12[9], and then subjected to both a qualitative and quantitative data analysis. Rather than using saturation as an iterative technique to determine final sample size, saturation here was gauged by a combination of *thematic* and *data saturation* (Saunders et al. 2018). The observed "informational redundancy" in the interviews suggests that saturation may have been reached (Francis et al. 2010).

The first step in the data analysis was a qualitative assessment of the interviews. Following Bryman's (2016) content analysis procedure, transcripts were carefully coded using a sequential inductive–deductive approach. This process produced four main themes and twelve subthemes. Main theme 1, *Social Factors,* captured a range of social expectations for community life broken down into six subthemes: *Community Activities (CA), Community Governance (GOV), Neighborhood Relationships (NHR), Shared Resources and Skills (SRS), Shared Values and Interests (SVI), as well as Social Support (SS).* Main theme 2, *Spatial/Infrastructural Factors,* tapped in interviewees' wishes for certain geospatial features and/or architectural characteristics and encompassed four unique subthemes: *Facilities, Services, and Other; Community Spaces (CS); Location;* and *Size.* Main Theme 3, *Negotiation of Privacy within Community* comprised two main concerns (also subthemes): *negotiation of time* and *negotiation of space.* Finally, main theme 4, *Miscellaneous Community Issues,* captured references to community that did not neatly fit into any of the other themes. Taken together, the themes provided an intimate look at the types of communities tiny housers viewed as ideal.

The interviews were then subjected to a quantification analysis to further elaborate and substantiate patterns seen in the qualitative portion of analysis that were suggestive of three unique *types* of community. A six-item *Community Engagement Style (CES) Index* was developed and interviewees' responses scored across six of the twelve subthemes (i.e., community activities, common spaces, shared resources and skills, social support, community governance, and meaningful relationships). This procedure produced six subtheme scores (ranging from 0–2) as well as a total community engagement score (ranging from 0–12) for each interviewee. Based on their total CES scores, participants were then placed into one of three groups believed to capture an underlying *community type.* Group 1 (a low-intensity CES indicative of a *mainstream view of community*): scores from 0 to 4, Group 2 (a moderate-intensity CES reflective of a *cohesive view of community*): scores from 5 to 8, and Group 3 (a high-intensity CES suggestive of a *collaborative view of community*): scores from 9 to 12. Finally, using IBM SPSS Version 24, a series of one-way ANOVAs (with Tukey HSD post hoc comparisons) were ran and applicable effect sizes $\eta^2$ calculated. The rationale for this analysis style was to determine if (and how) *low, moderate* and *high intensity CESs* differ on the six dimensions of community engagement and the overall CES index. Group differences were interpreted as further evidence for possible differences in types of desired communities. Assumptions for the statistical tests such as normality and homoscedasticity were checked and found to be generally met (Field 2018; Richardson 2011). By using a quantification approach alongside a qualitative content analysis, the researchers hoped that a more comprehensive picture of tiny housers' views of community would emerge.

---

[9]　NVivo is an advanced qualitative analysis software developed by QSRinternational, its offices are headquartered in Melbourne, Australia.

## 4. Results

In talking about their desired communities, participants touched on a range of topics including thoughts on what community means to them, reasons for wanting to live in community, as well as financial aspects of community life. The findings of this study suggest two overarching themes: an interest in specific community features and a perceived inevitability for having to safeguard privacy within community. When tiny housers talk about community characteristics, they either discuss desired *social aspects of community* (community activities, common spaces, shared resources and skills, social support, neighborhood relationships, community governance, and shared values and interests) or *spatial/infrastructural characteristics* (community spaces, facilities and services, location, and size). The findings also suggest that tiny housers' views of community may be best understood by examining how, and, in what specific ways, individuals seek to engage with other community members. Often couched in an unarticulated view of community (*mainstream, cohesive or collaborative* forms), their preferences show a desire for, more or less, distinctive *community engagement styles.* While all tiny housers appear to have explicit ideas of what the community should look like, most tiny housers, irrespective of their community engagement preferences, want to negotiate privacy in community (i.e., via regimenting time and space). To provide the reader with a better sense of the types of communities interviewees seek, the following section will sketch out a basic morphology of community characteristics and then address how interviewees negotiate privacy in community.

*4.1. General Community Characteristics*

The qualitative analysis of the interview data suggests tiny housers want to live in communities with seemingly similar spatial/infrastructural and social characteristics (see Table 1). All participants talk about wanting some form of *community spaces (100%)* or *community facilities and services (100%).* Many spoke about wanting a *place-based community of a certain size (83.3%)* and/or *specific geographic features (66.7%)*. Several tiny housers also presented clear ideas about what types of social interactions they want. They desire places that would allow them to, varying degrees, participate in *community activities (100%)*, draw on *shared resources or skills (75%)*, or benefit from *a commitment toward shared values and interests (87.5%)*. They seek communities with *social support (79.2%), good neighborly relationships (87.5%)*, and some form of *community governance (87.5%)*. These findings, while informative, potentially mask distinctive preference structures captured by different community engagement styles. Tiny housers with a low-intensity engagement style (CES), for example, hold less interest in the social aspects of community (see Table 1 for a comparison). They appear to echo a concern for *sharing values and interests* (83.3%), with those in the moderate-intensity (90%) or high-intensity CES (87%). However, to them, *community activities* (50%), *community governance* (66.7%), *neighborhood relations* (66.7%), *social support* (50%), or *sharing resources and skills* (16.7%) were less central. These findings are substantiated, in part, by the results of the one-way ANOVAs (see Table 2) which suggest that differences exist across CESs on four of the six dimensions (community activities, common spaces, shared resources and skills, and neighborhood relations). Tukey post hoc comparisons further indicate that tiny housers in medium and high-intensity CESs have similar expectations, except with regard to their views on common spaces (see Table 3).

*Spatial and Infrastructural Preferences:* Table 1 also suggests that interviewees prefer a particular set of spatial/infrastructural community characteristics. Most tiny housers (83.3%), for example, want to live in small to mid-sized communities (less than 100 houses). Other interviewees are less interested in the size of communities and instead stress the need for certain housing densities. More than half of interviewees also have clear expectations about the location of their community (66.7%).

They want homes in "rural" or "urban" settings with easy "access to the city," "green landscape[s]," or locations that are near "a body of water". Regardless of their geo-spatial aspirations, all tiny housers stress a desire to have community spaces, facilities, and services in their communities. While for some, this meant having "walking paths," a "pool," or "gym", for many this involved sharing spaces that

can facilitate higher levels of interaction, such as a "common center," a "communal kitchen," or a "community garden" (also see Table 1).

**Table 1.** Desired Community Characteristics Identified Via Interview Content Analysis [Broken Down by Overall Sample and Type of Community Engagement Style (CES) Group].

| | CES Groups | | | |
| --- | --- | --- | --- | --- |
| | Low | Moderate | High | Overall |
| **Social Factors** | 100.0% | 100.0% | 100.0% | 100.0% |
| Community Activities (CA) | 50.0% | 100.0% | 87.5% | 83.3% |
| Community Governance (GOV) | 66.7% | 90.0% | 100.0% | 87.5% |
| Neighborhood Relationships (NHR) | 66.7% | 90.0% | 100.0% | 87.5% |
| Shared Resources and Skills (SRS) | 16.7% | 90.0% | 100.0% | 75.0% |
| Shared Values and Interests | 83.3% | 90.0% | 87.5% | 87.5% |
| Social Support (SS) | 50.0% | 90.0% | 87.5% | 79.2% |
| **Spatial/Infrastructural Factors** | 100.0% | 100.0% | 100.0% | 100.0% |
| Facilities, and Services and Other | 66.7% | 80.0% | 75.0% | 75.0% |
| Community Spaces (CS) | 100.0% | 100.0% | 100.0% | 100.0% |
| Location | 50.0% | 90.0% | 50.0% | 66.7% |
| Size | 66.7% | 80.0% | 100.0% | 83.3% |
| **Negotiation of Private and Public** | 66.7% | 100.0% | 75.0% | 83.3% |
| Negotiating Space | 50.0% | 100.0% | 75.0% | 95.0% |
| Negotiating Time | 16.7% | 30.0% | 50.0% | 40.0% |
| Other | 16.7% | 40.0% | 0.0% | 25.0% |
| **Misc. Community Issues** | 100.0% | 100.0% | 100.0% | 100.0% |
| **N** | 6 | 10 | 8 | 24 |

**Note:** Shaded Characteristics Constitute the Facets of the Community Engagement Style Index.

**Table 2.** One-Way ANOVAs. Table shows between-group differences in community engagement across low, moderate, and high CES groups.

| | SS | df | MS | F | Sig | Eta$^2$ |
| --- | --- | --- | --- | --- | --- | --- |
| **CES Facet 1**: CA | 5.017 | 2 | 2.508 | 4.538 | 0.023 * | 0.302 |
| **CES Facet 2**: CS | 3.625 | 2 | 1.813 | 3.683 | 0.043 * | 0.260 |
| **CES Facet 3**: SRS | 6.125 | 2 | 3.063 | 5.937 | 0.009 * | 0.361 |
| **CES Facet 4**: SS | 1.750 | 2 | 0.875 | 1.208 | 0.319 | |
| **CES Facet 5**: NHR | 3.750 | 2 | 1.875 | 3.857 | 0.037 * | 0.269 |
| **CES Facet 6**: GOV | 3.725 | 2 | 1.863 | 3.230 | 0.060 | |
| **CES INDEX** | 147.792 | 2 | 73.896 | 105.506 | 0.000 * | 0.909 |

**Notes:** N =24, Community engagement style measured via CES facets that range from 0–2 [e.g., CA = Community Activities, CS = Common Spaces, SRS = Shared Resources and Skills, SS = Social Support, NHR = Neighborhood Relationship, GOV = Governance], * statistically significant at p=.05, Eta$^2$ effect size measure.

*General Social Preferences:* Interviewees also mention wanting to live in communities that offer community activities, community decision-making structures, and support for one another. They are looking for good neighborly relationships, shared community values and interests, and for some, opportunities to share resources and skills (see Table 1). Most tiny housers (87.5%) also report that sharing values and interests with other residents is important to them. They want others to value or take interest in things such as "tiny house living," "sustainability," or "diversity." The majority of participants (83.3%) also stress that they would like to have activities within their community ranging from events such as "potluck dinner[s]" and "movie night[s]," to more interactive "educational session[s]." Most tiny housers (87.5%) feel that they want to "get to know others" and "build relationships" with neighbors or live in communities that offer social support (79.2%). Interviewees also spoke about wanting reassurance that people would "check in" with them every once in a while, or "help" others, even with something as simple as "giv[ing] someone a ride." Sharing resources or skills, as diverse as "tools," "laundry machines," and "building knowledge" also ranked high on the wish list (75%).

Tiny housers also touch on community governance with 87.5% of participants expressing interest in either having a formalized system (e.g., "Homeowners Association (HOA)" or "Sociocracy"), and/or an informal mechanism to make decisions and resolve problems (e.g., forms of "self-governance"). While it may be tempting to equate these shared interests with a homogenous view of community, views of an ideal community do differ. To unravel some of these complexities, the following section will use both qualitative and quantitative evidence to suggest that a particular CES may coincide with a specific underlying view of community.

**Table 3.** Tukey Post Hoc Comparisons.

| | $M_{Low}$ (SD) | $M_{Mod}$ (SD) | $M_{High}$ (SD) | MD | SE | Sig |
|---|---|---|---|---|---|---|
| **CES Facet 1: CA** | | | | | | |
| Low–Moderate CES Groups | 0.33 (0.516) | 1.40 (0.699) | | −1.067 | 0.384 | 0.011 * |
| Moderate–High CES Groups | | 1.40 (0.699) | 1.38 (0.916) | 0.025 | 0.353 | 0.944 |
| Low–High CES Groups | 0.33 (0.516) | | 1.38 (0.916) | 01.042 | 0.402 | 0.017 * |
| **CES Facet 2: CS** | | | | | | |
| Low–Moderate CES Groups | 0.83 (0.753) | 1.00 (0.816) | | −0.167 | 0.362 | 0.650 |
| Moderate–High CES Groups | | 1.00 (0.816) | 1.75 (0.463) | −0.750 | 0.333 | 0.035 * |
| Low–High CES Groups | 0.83 (0.753) | | 1.75 (0.463) | −0.917 | 0.379 | 0.025 * |
| **CES Facet 3: SRS** | | | | | | |
| Low–Moderate CES Groups | 0.17 (0.408) | 1.00 (0.667) | | −0.833 | 0.371 | 0.036 * |
| Moderate–High CES Groups | | 1.00 (0.667) | 1.50 (0.926) | −0.500 | 0.341 | 0.157 |
| Low–High CES Groups | 0.17 (0.408) | | 1.50 (0.926) | −1.333 | 0.388 | 0.002 * |
| **CES Facet 5: NHR** | | | | | | |
| Low–Moderate CES Groups | 0.83 (0.983) | 1.50 (0.707) | | −0.667 | 0.360 | 0.078 |
| Moderate–High CES Groups | | 1.50 (0.707) | 1.88 (0.354) | −0.375 | 0.331 | 0.270 |
| Low–High CES Groups | 0.83 (0.983) | | 1.88 (0.354) | −1.042 | 0.377 | 0.012 * |

**Notes:** MLow . . . mean of low-intensity CES group, MModerate . . . mean of moderate-intensity CES group, MHigh . . . mean of high-intensity CES group, MD . . . mean difference between CES groups, SE . . . standard error, SD . . . standard deviation, * . . . statistically significant at $p = 0.05$.

### 4.2. Community Types: Reflection of Different Community Engagement Styles

Results from the one-way ANOVAs suggest that tiny housers hold similar expectations when it comes to the discussion of wanted social support and community governance. However, a closer examination of the data suggests that community engagement style is indicative of the desired types of community activities, common spaces, neighborhood relations, as well as shared resources and skills that tiny housers desire (see Table 2). Tiny housers with a low-intensity CES, for example, envision communities reminiscent of "mainstream" communities characterized by infrequent interactions among residents and only a basic social infrastructure[10]. Interviewees in the moderate or the high-intensity CES, on the other hand, are more likely to value communities with well-developed social infrastructures and/or widespread opportunities for human interactions ("cohesive" and "collaborative" communities). Post hoc comparisons of the different engagement style preferences also suggest that tiny housers with a moderate-intensity CES place a higher premium on more interactive community activities. In contrast, tiny housers in the high-intensity CES appear to see common spaces as more central to their vision of an ideal community. To them, common spaces constitute the principal means through which meaningful interactions among community members can be established and maintained (see Tables 1 and 3). In short, these three types of engagement preferences reflect tiny housers' implicit or explicit socio-cultural understandings of what community ought to be.

---

[10] Social infrastructure is a term that encompasses all community spaces capable of bringing people together, promote social interaction and encourage the formation of strong social ties. See (Klinenberg 2018).

### 4.2.1. "Mainstream" Views of Community [Low-Intensity CES Orientation]

Tiny housers with a low-intensity CES orientation seem to want communities more reminiscent of contemporary "mainstream" communities. However, they do share the other tiny housers' desire for some form of social support, some style of governance, as well as being around people that share their values and interests. Interviewees who endorse a low-intensity CES, however, less frequently state wanting specific community characteristics, and instead talk more generally, if at all, about wanting to have "shared spaces" or "events for people to do activities together". They also appear to be much less interested in being close to other residents, and thus seem to have a decidedly more passive view of common spaces, neighborhood relations, or community activities. And, to round it out, the promise of being able to share resources and skills with other community members, of high importance to tiny housers in other CESs, does not resonate with this group (for quantitative analysis see Tables 1–3). Rather than being a sign of rugged individualism, however, these community preferences may simply reflect "mainstream" views of place-based community; community which serves as a geographic placeholder where residents *by default* expect infrequent contact and are content with surface-type social interactions.

*Common Spaces:* Tiny housers in this group express a want for common spaces (see Table 1), though they generally seek common spaces less conducive to intense social interactions and are more often reflective of desired "services" (a "pool," "playground" for children, a "central farm," or "gym"). While some touch on more interactive types of common spaces, most comments lack elaboration. Ben, an exception in the low-intensity CES, "like[s] the *idea* of communal living" with a "central kitchen," or a "central farm" because it forces individuals out so they "actually . . . communicate." Like Ben, Mary admits that she definitely likes the idea of a "community center type thing" alongside with a "gym" and "pool" but does not elaborate on this idea. Venus was even clearer about her lack of interest in highly interactive community spaces. After being probed further as to whether she would want to see certain kinds of shared common spaces in her community, she simply replied "[n]o that's fine" citing her desire to remain "self-contained". Individuals in this group are also more likely to equate common spaces with spaces that provide public "services" and "activities" available to residents (e.g., concerts, pools or coin laundry rooms). Aware that other tiny housers love "community centers and community gardens," Artemis thinks that this "sounds really nice." She is aware that she and her partner could probably benefit from community gardens, and that she might be "most successful at growing anything . . . [if] there are other people who are knowledgeable" around, but her responses indicate a more abstract or hypothetical commitment to these spaces. Dan, an ardent disciple of simple living, also stresses spaces in terms of personal gain for his role as an "entertainer" and "personal trainer". He even imagines investing in a second tiny house where he could "do the training . . . with people in [his] own community." Common spaces to individuals in this group thus seem be more of an after-thought or a wouldn't-it-be-nice-to-have service or space, rather than a core component of their ideal community.

*Community Activities:* Another indication that individuals in this group are seemingly less concerned with establishing social relationships in their communities comes from their discussion about community activities, or lack thereof. While only half of those in the low-intensity CES even mention wanting activities such as a "cookout" or "watching movies," one individual, Venus, explicitly states she is "*not a joiner*". Others like Ben and Mary do express some interest in community activities, but their responses lack the enthusiasm and depth present in other CESs. Ben, for example, can see himself engaging with others and "*kind of* has a cookout plan" but fails to provide specifics. The hypothetical nature and the way he spoke about communal activities showed that high-intensity interactions are less important to him. Mary also claims to "love" the "community activities" such as "sports stuff," "music," and "movies." Her comments, however, follow a simple format and refer to things put on for the community (rather than being co-produced by its residents). With no mention at all, it appears, that for Dan community activities are also not at the forefront of his conception of community. Since most of these comments tend to revolve around entertainment activities commonly

organized by private or public entities, it appears that tiny housers in the low-intensity CES tend to seek activities frequently found in "mainstream" American communities.

*Neighborhood Relationships and Social Support:* The Tukey post-hoc comparisons between tiny housers with low and moderate-intensity CES seem to suggest no significant differences in the nature of desired neighborhood relationships. Looking at the group means, however, suggest that the pattern may simply be masked by the internal variation within the moderate-intensity group (see Table 3). The generally lower interest in neighborhood relationships among tiny housers in the low-intensity CES may further substantiate this; only 66.7% of interviewees in this group mentioned wanting good neighborhood relations as opposed to 90% in moderate and 100% of high-intensity CES members (see Table 1). When these interviewees spoke about neighborhood relationships, they mentioned wanting to build "bonds" with others, yet often failed to discuss how that might happen. Some spoke about "common interests" or "being outside more often" which could be seen as precursors for good neighborly relations and social support. Others stressed that they wanted to be with people for "fun" or to "share experiences," but many comments retained a more generic reasoning. Ben and Abraham, in contrast, express a stronger desire to be around others. Abraham, as the following quote indicates, wants people in his life but concedes that "transient" relationships maybe the best he can do. He puts it this way:

> "[That] kind of fills in that gap of being a single person at sixty-six and not having a partner, but being able to have people. Yeah, they're transient friends, but they're people that you know share the lifestyle and share the experience with you"—Abraham (age 55+)

Ben is highly aware of the pervasive loss of "connection to other people" but believes that this is something one can do little about. He wants to combat this "isolation" (and his own) by attending "game nights" or joining "Meetup" groups, but his examples—like those of other tiny housers in this group – only echo interactions reminiscent of place-based and liberated communities with lower levels of social cohesion. Venus, in addition to seeking good (mainstream) neighborly relations, would like to have access to more widespread forms of social support. She admits that "companionship with like-minded people" would be nice and hopes that in critical situations the community would step in and help a struggling resident. She said:

> "[I]f somebody got to the point where they were unable to pull their own weight [in the community] the energy investment that they had … will manifest itself by other members of that community taking care of that person." —Venus (age 55+)

She wants community members to know they "are not alone" but falls short of seeing social support as a "pre-given" in the community irrespective of the nature of the problem. Dan and Abraham also see "supportive" relationships in which people "help each other out" as important, but—like in other instances—talk about social support only in passing and in a more hypothetical fashion. Overall, the types of neighborhood relationships and social support that those in this low-intensity group seek tend to lack intensity, frequency, and substance. They abstractly agree on the value of these community characteristics, but their responses lack the depth, passion, and enthusiasm that tiny housers in other CESs express.

*Sharing Resources and Skills:* Tiny housers in this group, as both quantitative and qualitative analysis indicate, generally had little or no interest in sharing skills and resources with other residents (also see Tables 1 and 3). Venus, for example, commented that "yeah [sharing skills and resources] that's *fine*" but then moved on promptly to talk about something else she felt more passionate about. Participants' lackluster responses and general indifference toward the idea of sharing resources and skills suggest that those within the low CES do not view this community characteristic as a central component of their version of community.

While the data appears to suggest that tiny housers in this group want to build relationships with neighbors and have common spaces in their communities, their responses generally lack conviction,

depth, and elaboration. Common spaces that facilitate high quality and frequent interaction among residents may be nice in principle but are not critical to their overall conception of "community." In the same vein, these tiny housers generally value low-intensity activities in which residents are the recipients or beneficiaries of community-provided goods and services. They seek low-key and infrequent interaction with others in the community and are less likely touch on the reciprocal nature of social relations. Their views of community seem to reflect a slightly idealized version of the average urban, suburban, or rural American community, but they, nevertheless, still represent fundamentally mainstream connotations of community.

### 4.2.2. Cohesive Views of Community [Moderate-Intensity CES Orientation]

Tiny housers with a moderate CES want to live in more cohesive communities. While sharing many similarities with those in the high-intensity CES, they seem to value community activities and sharing skills and resources (see Table 3). They seek common spaces that are far more conducive to creating and maintaining strong interpersonal relations than those pictured by the low-intensity CES. Although some tiny housers express views on neighborly connections that share similarities to those in the lower-intensity group, most desire deeper, more fulfilling connections (more similar to those in the high-CES). They want community activities that allow them to "socialize" with other residents, and desire opportunities for skill and resource sharing like "tools" and "knowledge". While commitment depth may vary, there is little doubt that many seek cohesive communities to be around others so they can nurture deep meaningful relationships.

*Common Spaces:* While Tukey post-hoc comparisons of the ANOVAs seem to suggest that tiny housers in the in the low and moderate-intensity CESs want similar types of common spaces in their ideal communities, the in-group variations and qualitative data provide a more nuanced discussion (see Table 3). Tiny housers in this group have a better understanding and provide more illustrative examples of the role those spaces should serve within the community—even though it is less clear at times as to whether the primary goal of the space is a "service" (e.g., a "gym") or a principal means for connecting with others (a "clubhouse"). Tessa, for example, envisions her ideal community with a "pool," "dog park," or "internet café," suggesting a more utilitarian orientation. A more careful analysis, however, reveals her desire to "socialize" in these places. Like others, she recognizes that tiny houses inherently limit the "places to hang out [inside]." Basing her responses on her experience with "apartment living," she hopes to have community spaces like a "big T.V. room" in which others can share their love for movies. Others, such as Roselle, stress the importance of having common spaces that are useful *and* can provide interactional hubs for the community. This is especially visible in her desire to use a "shipping container" to house communal "washers and dryers," "showers," "a kitchen," and "food storage." Namor, in contrast, embraces the idea of certain "common areas" for the specific purpose of "gathering" and remains impartial to communal "kitchens" because they are too "personal" and pose a potential place of conflict over "cooking utensils;" she mentions that a "dining hall," "park," or "amphitheater" are fine. These and other comments indicate that tiny housers in this group want to use common spaces for members to "check on one another" or to simply "hang out" with another.

*Community Activities:* The Tukey post-hoc comparisons show that tiny housers in this group highly value interactive community activities, scoring no difference from those in the high-intensity CES (see Table 3). Table 1 further substantiates this observation. All individuals in the moderate-intensity CES (100%) want to have community activities. In comparison, only 50% in the low-intensity and 87.5% in the high-intensity CES do. To them, community activities are an important vehicle to foster deeper relationships. Samantha, for example, stresses that community activities are a "great way to meet people and share ideas and stories." She thinks that, "unless you have something to draw people out" you are "not going to meet everybody" and, as a result, will only know your "immediate neighbors." Tom, like Samantha also believes that as long as individuals can "choose" not to participate, activities can provide a powerful mechanism to "create a better sense of community," because it gives people a chance to "interact with each other." While these comments suggest a deep desire to connect, tiny

housers with a moderate-intensity CES differ in the types of activities they are looking for. Many talk about activities that express a strong willingness to come together not only for the sake of being "social" but in some instances to serve the greater good of the community. Ashley, Barbara, and Tessa, for example, all mention wanting "educational services" to spread knowledge, share information, or give people specific advice on "towing" and "building" tiny houses. Community activities can thus be seen as the primary nexus through which other community expectations can be addressed. By offering more than mere "entertain[ment]" or a chance to "socialize," they provide a structural base for "sharing" knowledge and building "bond[s]" with others in the community."

*Neighborhood Relationships and Social Support:* While tiny housers in the moderate-intensity CES have lower mean scores (though the Tukey post-hoc test shows no group differences) and mention neighborhood relationships slightly more often than their counterparts in the higher CES, they talk about them in a qualitatively different way. Most individuals in this group want to engage with others on a "daily" basis and develop a "sense of family". Greta, falling somewhere in the middle of the intensity spectrum, envisions a community where people "build a sense of "comradery" and "respect." While wanting everyone's "privacy" respected, she wants to live in a community where people do not feel "isolated" like people in the "suburbs". Others like Samantha unequivocally reject communities where people "don't care about each other" or are "so spread out that you don't know your neighbor". Ashley shares similar sentiments but rejects "current communities" for their "lack of [interaction]". Tom also loves deeply caring communities and doesn't want to be around those who are only "worried about how many initials are after [their] name on [their] business card." Interested in a "55 and up" community, Samantha has issues with a "busybody neighbor" but enjoys people who are "sincere" and really "car[e] about one another." She also does not mind having transitory or mobile people in her community, because she "loves to hear people's experiences" and wants to "live through them vicariously." Roselle echoes this selective emphasis when she stresses that she likes having *certain* people around, and that some people are just naturally a better "fit". While she considers her neighbors "friends" that she can "hang out [with] and talk [to]for like five hours" in one week, other weeks she is ok if they "just wave to each other in passing." She also stresses that she has no desire to "host" or "entertain," but wants to be seen as "a team player." These quotes and others like them show that these tiny housers seek qualitatively different neighborhood relationships and forms of social support.

*Sharing Resources and Skills:* Both the qualitative and quantitative data indicate that for those who desire to live in a cohesive community, sharing skills and resources comprises an important community characteristic. Compared to tiny housers in the high-intensity CES, individuals in this group are slightly less likely to mention wanting to share resources or skills (90% vs 100%). While their group mean is lower, Tukey post-hoc comparisons suggest that these tiny housers are as concerned about these highly collaborative community features as those scoring higher on the CES index (see Tables 1 and 3). They do not only want to share "normal" things such as "lawnmower[s]," "tools," or "communal chores," but some are also interested in sharing more "unconventional" things such as "clothes," "food," or insider "knowledge" (e.g., on how to properly "clean a composting toilet"). Jane, a dedicated tiny houser who likes the idea of sharing "pigs" and "goats," also envisions a community where people freely share knowledge. Jane yearns for people to share not because of "fees and dues," but rather, because "this is [their] community, [or] this is [their] family". While many want to share food from a garden, Roselle goes as far as wanting to "buy food in bulk and shar[e] it with everybody on the land." Others in this group hold more traditional views on sharing. Rachel for example, is "interested in [sharing]" and recognizes that there are "groups of people that do [share skills and resources]," but sharing is not something that is necessarily "at the forefront" of her mind. Tessa's idea of sharing also does not extend past a "wrench" and "a can of propane," expressing more reserved views of what should and should not be shared. Greta shows similar reservations. She says she has no desire to share her "blender," because she "uses it daily". She is ok, however, with letting things that people "use less frequently" such as "laundry [machines]" or "snow blowing [machines]" circulate in the community. Tom and Namor also have no objection that others share their simple tools like a

"shovel," a "hammer", or a "saw," though Namor is somewhat more hesitant to let others use his "nicer tools". He worries that they would not treat them with the same "respect" and fears that someone would even "tear [them} up." Despite these reservations, the quotes clearly illustrate that sharing resources and skills is a central theme for tiny housers in this group.

Taken together, the findings suggest that tiny housers who place interactive activities at the heart of communal life, express a much more cohesive view of community. They want to "bond" with others and build a real "sense of community." Although individuals in the group may differ in the strength and nature of their convictions, they share a basic desire for stronger relationships with other residents in their neighborhoods. Most of these communally minded tiny housers also consider sharing (even unconventional things) as central to their conception of community. While some of the comments may retain traces of mainstream thinking, most remarks show that they value some of the very same community characteristics embraced by those tiny housers that hold more collaborative views on community.

### 4.2.3. Collaborative Views of Community [High-Intensity CES Orientation]

Tiny housers that express a high-intensity CES have very high expectations for community engagement. They tend to seek communities that maximize the potential for interaction, collaboration, and strong relationships among residents, as well as stress how certain kinds of common spaces, neighborhood relationships, shared resources and skills, and community activities can achieve this. Looking at the Tukey post-hoc comparisons (see Table 3), what seems to set tiny housers with a collaborative view of community apart from other tiny housers are their views on common spaces. While their views on neighborhood relations or sharing resources and skills may not differ at a statistical level from tiny housers who embrace a cohesive view, the group means and qualitative data suggest these are very important concerns to them (see Table 1 for frequency counts). Through placing a premium on common spaces, they recognize—in one way or another—that having a well-developed social infrastructure is key. Common spaces such as community centers, communal kitchens, and/or community-based game rooms have the potential to encourage residents to engage in meaningful common activities that have important spillover effects into communal life. They perhaps allow community members to get to know one another, build trust, and develop a willingness to cooperate. This could not only improve neighborhood relations, but also motivate residents to return to these places and encourage them to share a wider range of resources and skills.

*Views on common spaces:* Tiny housers with a collaborative view of community were more likely to discuss common spaces and give extensive examples for them. They often passionately talked about wanting quality spaces in their communities that facilitate deeper and more meaningful engagement among residents. While they all agreed that having a "community garden" is highly appealing, many stressed communal dining facilities as an absolute must. Canan, for example, got really excited when he spoke about the potential of this. He imagines developing a community-run "food forest" where the community can grow its own food and use it to prepare daily "community meals". He also has an interest in creating a "makerspace" that can be easily reconfigured into a "dining room," a "classroom," or a "yoga studio". Mirroring Canan's passion, John discussed how he would like to "maximize [the social] infrastructure [so as to] . . . support [the] entertainment and educational needs" of the community. He sees "parks" as potential spaces for children and adults to congregate and envisions "a central pool" or "outside place" where community members are able to hang out or listen to music. He even wants to have a space that could be used to "tutor" someone struggling with "mathematics". Jenny went into equal detail describing her ideal communal spaces – most of which were places capable of facilitating high-quality engagement among residents. She especially favors a "community kitchen," "community library," or "community game room". Sebastian echoes Jenny's vision for a real "community kitchen" but would also like to see a "shared garden" and "music venue," because he thinks this could potentially help "develop a common interest" among community members. Examples such as these, and many like them, illustrate that tiny housers in the high-intensity

CES appear to be quite serious about having shared community spaces that can facilitate genuine human encounters and provide spatial context for fun community activities to occur.

*Community Activities:* Like those with a moderate CES, tiny housers in this group emphasize the importance of highly interactive community activities. They desire activities that allow them to "engage with others" and give them the freedom to enjoy a wide range of "entertainment." Tim, for example, wants a schedule of activities that allows "people [to] engage on a regular basis." By talking about activities that range from frequent music and poetry "festivals" to communal "dinners," he echoes Canan's emphasis on structured activities. Canan does, however, go a step further and sees much room for highly-interactive activities such as "study groups" where people could discuss topics they are passionate about, alongside frequent "community meals" where members are required to "help prepare the food . . . [and then] maybe have fun . . . playing a game [together later]." Others, like John and Rick, expressed interest in opening up their homes and turning them into semi-public or public spaces. John loves "entertaining" and "wants [to have] groups over and cook out." Rick shares these sentiments and says that "sharing meals together" at his Le Petite Domicile is a great starting point. Rick's ultimate dream, however, is to live in a community where people are "less stressed" and "more willing to connect" – in what he calls "a deep way." He wants to "do stuff together" from having "bonfires" with other residents to "study[ing] the bible". Archie, on the other hand, has no real preference when it came to activities. Simply knowing that enough "opportunities [to engage] are there" was "most important" for him. While most of these comments highlight a deep-seated desire for high-quality interpersonal relations, many tiny housers in this category also seem to want to use activities for "outreach" purposes. They felt strongly that communities should not stay isolated, but rather they needed to remain connected to the wider community. Nancy, for instance, talked about having "open tours" where people from outside the community can visit and learn more about the beauty of tiny house living. Like other interviewees, she loves the idea of "giving back." She spoke about wanting to create "street paintings" or "build public benches and a seating area out of cob," which can serve the community at large. When looking at the comments on community activities, it becomes obvious that these tiny housers care passionately about activities that connect people both inside and outside of the community in deep and meaningful ways.

*Neighborhood Relationships and Social Support:* The results of the one-way ANOVAs and the Tukey post-hoc comparisons seem to suggest that tiny housers with a moderate and high-intensity CES do not differ on their views on neighborhood relations and social support (see Tables 2 and 3). The qualitative data, however, paints a more nuanced picture, which seems to indicate that tiny housers in this category express a consistently deeper desire for communities that foster "authentic" "companionship," often citing a lack of neighborliness and social support in mainstream communities. Many advance narratives highlighting a desire to "get to know" their neighbors and "support" them as best as they can. Jenny, for example, believes that most "people hardly spend [any] time with their neighbors", and that "real" communities are "greatly lacking in this country, because we isolate ourselves" nowadays. The following quote illustrates this wonderfully:

> "When you look at most neighborhoods, people come into their house, they drive into their garage . . . they walk into their homes from their garages, they spend the majority of the time either at work, at home or at various things outside of their home . . . and they hardly ever know their neighbors"—Jenny (age 35–54)

John seconds these views by saying that "more and more [people] in the United States . . . have withdrawn, and front porches have disappeared, and [instead] fenced backyards have appeared" everywhere. Likewise, Sebastian exclaims, "people are isolated from each other" today. He believes that people who live "in suburban neighborhoods where people have garages and busy lives that they are slaves to" are especially susceptible to "isolation." In an effort to combat the problem that "people don't take the time to care about their neighbors" anymore, he wants to use "intelligent design" elements that could "increase the likelihood of interactivity." Rick, like others in the high-intensity

CES, values "openness" among neighbors, and believes that "relationships [can] form from having get togethers" or "dinner" at his own home. He hopes to spend time "daily with people" - people "from all different types of backgrounds". Tim, too, looks for

> "an environment where [he is] constantly being engaged with other people of differing ideas and [exposed] to various religious and cultural backgrounds [where] you share your experiences with [each other]"—Tim (age 55+)

To him, this diversity and sharing of ideas broadens ones "concept of community and what it means to engage with people in a positive way," which leads to a simple but beautiful "you help them and they help you" default mode for community interactions. Nancy shares Tim's predilections for highly cohesive and collaborative communities, but more strongly emphasizes a need for genuine "support networks." She eloquently puts it this way:

> "[community] is company when I want company. You know friends, when I want to hang out with friends, and when I don't want to, I can just go . . . [to the] house. It's knowing that if I am too exhausted to cook that probably somebody has a meal for me. [And] I have a place for somebody to come if they are having a bad day and they wanna have a beer or a cup of tea. They know that they [can] come and sit on my porch and I'll be there kinda thing. Or if my car breaks down or if I get a flat tire on my bicycle, there's a group of people that I can call on, and, likewise, they can call me if they have something heavy they have to lift or move, or if they need a tree cut down or whatever it is. It's like hey I'm here I can help you, what is it that you need, let's go [and] do it."—Nancy (age 35–54)

Powerful quotes like these, and many others, illustrate that tiny housers in this category are seemingly all about nurturing genuine connections with neighbors and/or creating communities where everyone exhibits a manner of just "being there for one another." This attitude and deep commitment to the "other" not only creates community with a capital "C" but also establishes the necessary conditions under which people are more likely to share resources and skills.

*Sharing Resources and Skills:* Tiny housers with a high CES have a seemingly unparalleled vision of sharing various resources and skills among the community. While the Tukey post-hoc comparisons did not detect differences between the moderate and the high group, the interview data and the group means clearly support this assertion (see Table 3). While tiny housers in this category want to share common items such as "lawnmowers" or "bicycles," many also believe that sharing can be used to do good in the community (a view that tiny housers from less intense CES's rarely or never expressed), in their views, as an opportunity to "support" people's various needs. Food, grown individually or communally, can be shared during "community meals," helping those that may be going through hard times. Jenny wants people in the community to share "leftover building materials," rather than them "just wasting away." Nancy embraces the—for some—unorthodox idea of wanting to buy a "52 pack of toilet paper at Costco" and suggests residents could "split that up and share it". She, like many others, was also a vocal advocate for a community "barter" or "trading" system which could allow the community to transcend the "normal" monetary system. She says:

> "I might cook meals for you if you come over and fix my [broken] outlet, or I'm a photographer and I'll do photos for an event that you're doing if you'll update my website."—Nancy (age 35–54)

Similarly, Archie wants to see "barters in the community" and loves the idea of "trading off" things in basic reciprocity schemes. Trades, too for him, could look as simple as someone exchanging their "video editing skills for a massage or . . . fruits from their garden . . . [in exchange for] help put[ting] solar panels on people's houses". Rick also respects these time-proven systems of exchange. He says that they are great precisely because they are "not so focused on . . . well I'll do this service

for you for this price [but more like] . . . well if you help me with this, I'll help you with that kinda thing." He strongly believes that this is "a better way of life because you're connecting with people at that point." John goes a step further and stresses that "sharing" should be thought of as a "service" to the community available to everyone "free of charge". Echoing the well-worn insights of the ABCD tradition, he wants to "inventory" the resources and skills every individual has because he truly believes that knowing who is a "mechanic" or "carpenter" in the community or who would be willing to offer "childcare" or "mentoring" services would be fantastic. There is a common thread that appears to run through most of these comments: these are highly community-oriented tiny housers that see sharing as something more than a simple utilitarian exchange. To them sharing seems to be a part of the deep connective tissue that binds members in collaborative communities.

Tiny housers that adhere to a high-intensity CES want to live in highly collaborative communities. They generally appear to gravitate toward communities with well-developed social infrastructure (i.e., common spaces), highly interactive community activities, cohesive neighborhood relationships, strong reciprocal social support systems, and non-mainstream approaches to sharing resources and skills. However, like the tiny housers within the other CESs, they also touch on a series of privacy concerns.

### 4.3. Negotiation of Privacy in Community

Most interviewees (83.3%) expressed a strong desire to create and maintain a sense of privacy in their desired communities. While these concerns are most prevalent among tiny housers with a *cohesive* view of community (100%), those that lean more toward *mainstream* (66.7%) or *collaborative* community ideal types (75%) also touched on this theme (see Table 1). Privacy concerns generally revolved around issues of space (95%), time (40%), or wanting autonomy with respect to decisions and/or resources (25%). These recurring themes suggest a practical concern for privacy that can potentially arise from the realities of living tiny as well as demonstrate the pervasiveness of an individualistic socialization.

### 4.3.1. Negotiating Space

Most tiny housers who mentioned privacy concerns wanted to ensure that they would have their own "individual space" in some form or another (95%). While some spoke about wanting a "plot of land" to themselves, many worried about the "amount of land between homes," especially being "cramped . . . next to the other," or potentially having to live "on top of each other." Tiny housers also stressed the importance of visual, physical, and normative barriers in their communities such as "trees," "[window] blinds", or set rules that help to reinforce spatial privacy. While these themes run through all community engagement styles, those in the low (50%) and high-intensity categories (75%) appeared slightly less concerned than individuals from the moderate-intensity group (100%). Irrespective of their views on community, however, interviewees stated or implied a desire for communities with "smart design[s]" that address these issues.

Most tiny housers who express an uneasiness about spatial privacy in community talk about plot size, spacing issues, and/or the design decisions that go into this. Nancy, a middle-aged woman, for example, contends that "just about every tiny houser" she knows is concerned about how communities are designed. She argues that the financial interests of planners and developers often overshadow residents' needs. To her, communities and residential neighborhoods are built to maximize physical space, sidelining concerns for individual privacy in the process. Nancy used the following metaphor to convey her disagreement with this approach:

> "[M]ost developers, you know . . . are all about the money . . . they want to cram as many [houses] into a space just like an RV park or a mobile home park. They want to line 'em up all pretty like dominoes and cram as many in there as they want . . . [but] tiny housers want it to be more like a game of pickup sticks where you would just scatter a handful of wooden matches and that's how they land and that's how they want their house planted. They don't want to be on top of each other in a really organized orderly fashion"—Nancy (age 35–54)

While not everyone touched on the developer and building side of the equation, most tiny housers were clear in their wishes for proper spacing among homes to safeguard their privacy. Struggling with the demands of community and the desire for community, they often talked about wanting "enough space for privacy yet community." Mary, like others, also acknowledges that social connectedness and privacy can coexist as long as community members share similar values. To her

> "It might even be better to have more private space in between the home, but to have a network of people who are trying to create the same sort of community."—Mary (age 55+)

Talk of coexistence, however, was often overshadowed by a strong desire for "personal space." This desire was often couched in elaborate statements of not wanting to "see," "touch," or "hear" their neighbors. Some tiny housers provided specific distances that they felt were "adequate." Tess, for example, felt that a buffer of "ten feet" between homes would suffice. Others made references to the proper "sitting" and "orientation" of the tiny house which underscores these concerns.

Interviewees also discussed specific plot sizes that would suit their privacy needs. Namor, for instance, thought that adequate property sizes could help create appropriate spacing to his neighbors. Rather than compromising on size, he wants "the same plot of land that a regular house is built on" to give him "more room to do with as [he] please[s]." Others, like Tessa, don't particularly "care for yards," they just want enough room to not feel "crowded." Many tiny housers, however, had a specific number in mind when it came to the ideal size of a property. While Barbara and Samantha think "an acre" is enough, Dan could see himself living on a "quarter of an acre," and Tim, who takes his minimalist beliefs to their logical conclusions, states that "a fifth of an acre" would probably do.

Comments on the adequate spacing of houses and "correct" property sizes predominated the interviews though discussions on visual, physical, and normative boundaries that help reinforce privacy came in a close second. Tiny housers wanted visual and physical boundaries such as "tree[s]," "fences," "[window] blinds," or spoke about the correct "positioning" of windows so that they are not facing other windows. Normative boundaries, i.e., the wish to have clear rules in place to protect private space, was another common theme. Roselle, a young mother of three who lives in a TH community, nicely captures all three aspects (visual, physical, and normative boundaries). She spoke about how these boundaries helped to maintain a sense of privacy and ownership of her space.

> "We all have our own fences that divide our spaces, and I think that out of all the communal living that I've done, I feel like living with fewer people on a piece of land has been like a saving grace … because any time there's a problem it's like okay you stay on your side of the fence, keep your stuff on your side and I'll keep my stuff on my side. You get to do whatever you want on your side and I get to do whatever I want on my side … ."—Roselle (age 18–34)

While having their "own fences that divide [their] spaces" serves as a visual and physical boundary, declaring her "own kitchen" and "laundry [machines]" as exclusive personal domains serves as a normative boundary to preserve her version of privacy. Greta, a middle-aged woman who likes being able to "see [neighbors] easily throughout the week" also desires a "sense of quiet and privacy." She believes that effective positioning of houses could offer her privacy and prevent her from "encroach[ing] … on other peoples' space." She put it this way:

> "I wouldn't want … every window … facing another window. That's one of the worst parts about urban living. It's like you can wave to and pass by, but not where you're in each other's business. Whether … you're having a fight … . you have kids up at two a.m. teething or needing a bottle, or … dogs. I love animals, [but] I don't want … dogs barking in the middle of the night or in the middle of the day or whatever."—Greta (age 35–54)

Taken together, these quotes illustrate that many tiny housers, despite their often-strong interests in community, also seem to entertain legitimate concerns about wanting to maintain their "own

space". These concerns may simply reflect the practical challenges that tiny living poses to privacy, or they might reveal aspects of their more individualist American socialization. While spatial concerns dominated the comments, some interviewees also stressed that they wanted control over their free time, or as Jane put it, time to enjoy their "own individual lives."

### 4.3.2. Negotiating Time Demands

Some interviewees expressed that they wanted to have enough time for themselves (40% of those who mentioned privacy concerns), though this "desire" was often mediated by their views on community. Tiny housers with views of community that place more demands on the individual socially generally expressed more concern for preserving time for "their own individual lives." While half of the tiny housers with a *collaborative* view of community (50%), for example, touched on this, only less than one third from the *cohesive* view did (30%). Tiny housers who envisioned living in a *mainstream*-style community were even less likely to mention this with only one interviewee hinting at this. This pattern suggests that the more important social interactions are to tiny housers, the more they seem concerned with the "time factor" (also see Table 1). Stated another way, most interviewees agreed that rather than being an "obligation," participation in community should be voluntary. To them, this means that each individual should have the "autonomy" or "permission to be alone", as illustrated by Roselle in the following quote:

> "I like to go to the communal spaces and socialize, but to be able to go and do my own thing in my own space . . . . That's so important. You have to have personal time. You can't just invest all your time in people or your balance is really bad."—Roselle (age 18–34)

Describing his "typical day," Canan, a tiny houser in his early sixties, shares some of Roselle's sentiments yet focuses more heavily on the cyclical nature of having to balance privacy and community demands. He expresses value in both having time to himself and staying connected to others. The following quote that highlights his position:

> "So a typical day would be being in silence for the first part of the day, then having a having a chat with people around breakfast time, going back to my tiny house, doing some more work and then maybe come again for lunch . . . then go back to work again. So, that's how I work best is by myself, but also not feeling as if I'm too disconnected from people. I can go talk to them when I want to. So that that would be a perfect day for me."—Canan (age 55+)

Venus, a self-labeled "extreme introvert," shares some of the same views. While she recognizes the importance of community, she also wants to be able to control which social interactions to participate in or not. Her interview made it very clear that, for her, it is less about juggling personal and community time and more about the quality of interactions. She hopes to be part of a community that would respect her decisions, while at the same time would give her a reason to leave the house. The following quote nicely captures her attitude:

> "If they're nice people, I'd like to go and visit. That's about it. Or, if people come together to accomplish a particular project, I'd be more than happy to participate . . . But I'm not a small talk person . . . I don't want to just sit around and you know, chew the fat. It has to be of substance."—Venus (age 55+)

Quotes like these, as well as others, show that while most tiny housers want communities that allow members to connect and support one another, many are keenly aware that privacy might become an issue. Interestingly enough, those who imagine themselves being more involved in the community express privacy concerns more often, suggesting some interesting interactive effects.

## 5. Discussion

The tiny house movement often invokes Thoreauvian visions of rugged individualists pursuing lives of self-reliance away from centers of civilization (Anson 2014, 2017). However, rather than being this misanthropic recluse suspicious of human relationships and society (Miller 2017), Thoreau was also someone who nurtured a deep curiosity, love, and care for others, even for those that society usually pushes aside as "outsiders, unknowns, [and] outcasts" (Kaag and Martin 2017; Moller 1980). His countercultural philosophy challenged many taken-for-granted realities of his time and helped cultivate a much more holistic understanding of community, one that weaves human beings into the complex fabric of the natural environment (Anson 2017; Cafaro 2001; Moller 1980). The findings of this study suggest that at least some of Thoreau's contemporary devotees, and possibly a larger cross-section of the larger movement, continue to honor Thoreau's commitment to "unconventional" forms of community. They have not only inherited his deep conviction that "we [all] belong to community" (Thoreau 2006), but have also adopted the belief of their "patron saint" that tiny living "cultivates [if not necessitates] a greater degree of reliance on neighbors, community resources, and shared land" (Anson 2018, pp. 334–35). While contemporary tiny house bloggers and tiny house TV shows have managed to reawaken a nostalgia for American frontier life among their receptive 21st century audiences, and with it, have created a vocal cluster of tiny house converts (Penfold et al. 2018), this more individualistic current within the TH movement may represent a minority. The findings of this study suggest that a segment of community-minded tiny housers want to be part of more cohesive and collaborative communities.[11] This is in line with previous findings by Harris (2018) who argues that many tiny housers strive for communities that are prefigurative, collaborative, and to some degree that "challeng[e] the status quo" (p. 70). While staking claim to a modicum of privacy, these heirs of simple living want to join small vibrant place-based communities that allow them to nurture deep meaningful relationships with friends and neighbors. Like Thoreau, they seek communities that defy the status quo of their day. They desire "unconventional" communities that allow people to spend time in common spaces, join interactive social activities, and share resources as well as skills (Shearer and Burton 2018). However, why do these tiny housers feel this way in a world that has moved toward liberated and post-place communities? What makes tiny housers articulate views of community that seem to run so counter to our globalized and cosmopolitan notions of modernity? Why do a large percentage of them express concerns over privacy while at the same time desire to live in such different communities? While these are difficult questions to answer, the following discussion suggests that these views may be best explained by references to the broader socio-cultural context of our time.

### 5.1. Desire for More Cohesive or Collaborative Community Structures

Tiny housers, like many other Americans today, spend much of their daily lives trying to navigate the powerful existentialist waves of our hyper-consumerist, deeply fragmented, and constantly changing lives (Giddens 1991; Ritzer and Dean 2015). In their efforts to regain control over their lives, many tiny housers strive for financial independence, personal autonomy, and new experiences. They also seek deep meaningful relationships and, by extension, community (Jebbink 2019; Mangold and Zschau 2019; Shearer and Burton 2018). Those who heed the calling for a simpler life, however, do not just settle for any type; they appear to prefer more cohesive and collaborative community structures. This quest for specific communal characteristics and modes of living, while complex, may simply be a part of their much longer journey going tiny. It may be through this process of introspection, that

---

[11] This argument seems to be borne out by the preliminary results from a large US-based survey of tiny housers (N = 453). These findings suggest that for over two thirds of tiny housers "living in community" is at least "somewhat" important (with 40.9% of them feeling strongly or very strongly about it). Tiny housers also seem to be interested in more collaborative community characteristics (e.g., as shown in their interest in wanting to share resources and skills, have designated common spaces in their communities or wanting to spend time with other residents). Depending on the particular community aspect and the expressed strength of interest, percentages here range from 40% to 80% (Authors, Unpublished Manuscript).

tiny housers become slowly exposed to ideas of other forms of communal living, which, over time, may change and/or refine their views on community. However, these preferences may also signal that theoretical arguments advanced by the "saved" and "liberated" perspectives on community have limited conceptual utility (Beck 1994, 2000; Wellman 1979; Wellman and Haythornthwaite 2002). The desire to return to strong place-based communities, which people in *this* segment of the TH movement express, may be better understood by synthesizing insights from the both perspectives (Goe and Noonan 2007; Putnam 2000).

Many tiny housers may seek these "alternative" forms of community because they feel a general sense of erosion of their place-based communities and/or see life in the newer saved, liberated, and post-place communities equally unfulfilling. Consistent with community lost accounts, this reasoning is supported by studies which have shown that place-based communities have vanished, eroded, or been transformed into communities that transcend time and space (Giddens 1990). Scholars offer a range of evidence for the "community lost" thesis and the negative psychosocial consequences that come with it (Popenoe 2001). Like a large segment of the US population, tiny housers are increasingly living in neighborhoods highly segregated by race, income, and social class (Thal 2017). Trust in communities and neighborly relations, especially in urban and suburban settings, has declined (Garoon et al. 2016; Yeo and Green 2017), and average network sizes, while remaining stable across different age groups (Smith et al. 2018), have also shrunk over the past few decades. Social isolation remains common (Klinenberg 2018), especially among older individuals who are at higher risk due to "living alone, being widowed, poor health, psychological distress, economic deprivation, poor social skills, and cognitive impairment" (Parigi and Henson 2014). Seen from this perspective, this may mean that tiny housers, like many of their American contemporaries, find it increasingly difficult to have conversations about "important matters," receive social support in times of need, and/or have networks that have deep meaningful relationships. (McPherson et al. 2006)[12].

Research echoing the "community liberated" (or post-place community) tradition, on the other hand, suggests that rather than "losing community," Americans are managing to find "connection" elsewhere. They increasingly build post-place communities with "larger and more diverse core networks" which tend to create a "pervasive awareness" of the "other" (Hampton et al. 2011). These communities, even those exclusively virtual, seem to emulate a sense of community that offers members a similar range of cognitive, social/personal integrative, as well as hedonic benefits as place-based communities (Tonteri et al. 2011). The fact that tiny housers in this study spent little to no time talking about aspects of liberated and post-place communities, however, may indicate that these places fulfill more of instrumental functions for them. Like voluntary simplifiers, they may use these communities to exchange information, have fun, and/or coordinate larger events. It seems, however, that many tiny housers see those virtual networks as inadequate psychosocial substitutes for the communities they truly seek (Huneke 2005; Katra 2017). While it remains unclear to what degree non-place-based (or virtual) ties can truly "bind" participants (Turkle 2011), studies have hinted that virtual communities are most satisfying when they co-evolve with or turn into place-based communities (Blanchard and Horan 2000).

Whether liberated communities can offer the same (or better) experiences and psychosocial outcomes as their place-based cousins remains highly contested. What matters, however, is not the objective but the *perceived* state of our communities. Tiny housers that embrace alternative visions of community, it is argued here, may simply feel discontent with the quality of their own communal lives. A true "sense of community," as (McMillan and Chavis 1986) have argued eloquently, requires individuals to have access to four key community elements: membership, influence, integration and fulfillment of needs, and a shared emotional connection. More community-oriented tiny housers may

---

[12]　The findings of McPherson et al. (1909) are a topic of hot debate within sociology. For further information about these findings please see (Fischer 2009; Hampton et al. 2011).

simply feel that one or more of these elements is missing from their own lives leaving them unfulfilled. They may feel an adequate sense of community but come to learn that other community-based lifestyles are equally or more appealing. With its focus on physical place, shared communal spaces/activities/resources, and active community involvement, this part of the tiny house movement seems to share meanings of community with the co-housing or ecovillage traditions. Co-housing principles, for example, advance a Feng Shui-like philosophy that stresses how spatial characteristics of buildings, outdoor spaces, and overall physical design can elevate communal energy levels and encourage social interaction. By striking a careful balance between individual needs and the demands of the collective, many cohousing initiatives and ecovillages hope to create collaborative communities that offer residents the best of both worlds. These two models also offer non-conventional blueprints that fundamentally challenge Western connotations of home, community, and sustainability (Sargisson 2012; Van Schyndel Kasper 2008; Williams 2005). Tiny housers attracted to these types of communities, however, seem to differ from voluntary simplifiers who tend to place more focus on self-development, often live in mainstream communities, and generally relate with fellow devotees in post-place VS circles (Huneke 2005). While many neo-Thoreauvian tiny housers have inherited the intellectual legacy of Walden, and with it, Thoreau's non-mainstream views of community (Anson 2018; Moller 1980), the contours of their ideal communities come in a variety of different flavors. This may not be all that surprising given that the philosophical orbits of the TH movement, intentional communities, and other intentional communities increasingly intersect.

Tiny housers are increasingly being exposed to alternative forms of communal living online. It may be precisely these ideas that have begun to challenge and alter their views of community. While prior studies have demonstrated that community is an important motivator for living tiny (Boeckermann et al. 2019; Jebbink 2019; Mangold and Zschau 2019; Mutter 2013), little is known about where tiny housers ideas surrounding community originate. Some scholars have suggested that since the tiny house movement remains primarily an online phenomenon (Hutchinson 2016), tiny housers derive much, if not most, of their information from blogs, news articles, and web-based forums (Katra 2017). Although many of these sources provide merely building advice, legal information, and/or stories of success and failure, there is an increasing number of blogs or websites dedicated to issues of community. Websites such as *tinyhouseblog.com* (Odom 2014), *tinyhouseexpedition.com* (Stephens 2019), or *tinyhousecommunity.com* (Walker 2019), for example, have become a new virtual mecca for individuals interested in joining or starting their own TH communities (Doman 2016). These resources provide tiny house enthusiasts with an interactive platform to discuss philosophical, practical, and at times rhetorical questions about community. It also gives them instant access to a wide range of information about different (often non-mainstream) approaches to community such as cohousing (Walker 2019), ecovillages (Bayly 2016), pocket neighborhoods (Grimm 2013), and permaculture (Alexis and Parsons 2018).

The exposure to other forms of community (e.g., cohousing and ecovillages) and living (e.g., minimalism) also seem to occur within face-to-face settings. Regional tiny house festivals, cohousing conferences, and workshops offered by non-profits (e.g., Atlanta-based MicroLife Institute) have become popular hangouts for veterans and novices to share their passion for alternative forms of community life (Kanto 2019, Microlife Institute 2019). As a result, these spaces have gradually become inundated with talk about walkable communities, co-housing principles, ecological footprints, land-trusts, permaculture, and/or communal spaces (United Tiny House Association 2018; Vercillo 2019)—suggesting a free flow of ideas across movement boundaries. Tiny housers are also increasingly exposed to minimalism's "less is more" philosophy, appropriating a range of minimalist ideas and terminology. Life goals are frequently expressed using the language of "decluttering," "simplifying," or "intentionality", and are modulated onto narratives that stress individual "happiness," the "good life," or "experiences" (Tiny House Blog 2019; Tiny House Expedition 2020). For more community-oriented tiny housers, these minimalist themes often mutate into a strong desire for social connection with their fellow human beings. Given the prominence of community narratives in movement spaces, one

seemingly dominated by alternative frames, it may not be all that surprising that individuals in this part of the tiny house universe have begun to embrace views of community that are more cohesive and collaborative.

*5.2. Maintaining Privacy in Community*

While many tiny housers appear to articulate a strong interest in cohesive and collaborative community dynamics, their comments also suggest that privacy is not only a precondition for community (Tunick 2001), but that privacy and community cannot easily be separated (Boone 1983). Tiny housers demand "spatial boundaries" while being intensely aware of their own "dependence on community resources such as storage facilities, showers, and garden space" (Anson 2018). Studies on residential privacy further empirically substantiate these claims. Individuals who feel their privacy is respected score higher on measures for sense of community (Wilson and Baldassare 1996) and personal well-being (Farrell et al. 2004). The general impulse of wanting to safeguard privacy while also wanting to secure access to community activities, shared spaces, or meaningful relationships, may thus be completely "natural." Tiny housers' specific concerns over personal time and space, however, may be simply due to the practical aspects of tiny living or their hyper-individualized socialization in American culture.

Tiny housers may express privacy concerns because their houses pose unique functional and psychosocial challenges. The homes of European upper and middle-class families in the 18th century—which can be seen as an architectural precursor of contemporary American houses—became spaces in which the public and private sphere converged. In his seminal work, the *Structural Transformation of the Public Sphere*, Jürgen Habermas (1991) argues that the "family room" (or living room) "became a reception room in which private people gathered to form a public" to discuss art, literature, and philosophy (p. 45). While they fulfilled a real political purpose at the time, these spaces (and others) have become "pseudo-public spaces" or repurposed "private spaces" that allow people today to spend much of their time indoors (Habermas 1991; Hutchinson 2016)[13]. Tiny houses, in contrast, eliminate these rooms from their floorplans and employ sophisticated multi-functional designs to maximize indoor space. These design decisions move private activities that usually occur within the orbits of specifically designated rooms (e.g., recreation rooms, laundry rooms, media rooms, and playrooms) and relegate them to the outside. Lacking the performative functions of the average home to accommodate public functions, tiny homes force their inhabitants to move aspects of their social life into the open. Add to the equation that tiny living frees up time maintaining the home, and it becomes clear that this architecture of simplicity can create strong psychological incentives for people to spend more of their lives outside the home.

Hutchinson (2016) in a brilliant discussion of tiny housers' spatial authenticity, for example, argues that "[s]maller, more private homes require better public spaces to compliment them" (p. 126). He maintains that the functional inversion of where activities now take place necessitates that people spend more "time . . . together" (Ibid., p. 114). The spatial imperatives of tiny houses and their behavioral consequences lead to a stronger need to balance community and privacy considerations (Brown 2011). [T]iny living "requires better private space" (Levin 2012) and a much "more careful consideration of privacy" (Hutchinson 2016). People—not just tiny housers—feel an urge to withdraw from social interaction when they perceive their privacy is being invaded or encroached upon (Baum 1977; Ravetz 1988). Studies on residential designs have demonstrated that critical dwelling densities exist which can help avoid overcrowding (Altman 1975). To maintain a balance between private and public domains, planners often use buffer zones (semi-private spaces) or public (or communal) spaces to maintain good neighborly relations. Semi-private space or buffer zones (gardens and verandas,

---

[13] The National Human Activity Pattern Survey (NHAPS), for example, found that the average American spends close to 70% of their daily time in their homes (see Klepeis et al. 2001).

etc.) also have been used to create gentle transitions between public and private space (Abu-Ghazzeh 1999) and thus provide "a degree of privacy and territorial control with options for active contact into adjacent public space[s]" (Skjaeveland et al. 1996). Other design features such as doors, walls, and windows alongside with curtains, screens, and visual divisions, as classic sociologists like Simmel (1909) have argued, not only offer functional means to safeguard privacy but also carry deep cultural and symbolical meanings with them (Schwartz 1968; Simmel 1909). In short, the privacy concerns that many tiny housers have expressed may reflect an intuitive understanding of the social psychology of space and the sociology of residential areas. Their preoccupation with privacy, however, may also be indicative of their unique social and cultural upbringing.

Tiny housers, like their mainstream counterparts, are likely to grow up internalizing a specific set of cultural meanings about privacy and autonomy. It may be those taken-for-granted meanings that have shaped tiny housers' privacy concerns. While concerns over privacy are universal (Acquisti et al. 2015), cultures often differ widely on how they define and enforce privacy (Hall 1966, Nakada and Tamura 2005). Pre-modern cultures, for example, often have comparatively underdeveloped systems of privacy. Some tribal societies produce highly sophisticated social surveillance mechanisms that de-facto create an all-pervasive public from which one can only escape by leaving the community altogether (Altman 1977)[14]. Western conceptions, in contrast, focus more heavily on the individual, while Eastern and African perspectives stress community and the negative consequences of privacy more strongly Capurro (2005)[15].

The American understanding of privacy draws strongly on Western notions of spatial and behavioral autonomy as well as the idea of personal dignity (Laufer and Wolfe 1977). The contemporary social and cultural system uses privacy to structure micro spaces to facilitate the "conduct of daily transactions, the organization of space in . . . houses and buildings and ultimately the layout of . . . [entire] towns" (Hall 1966). They also offer intricate cultural ciphers for social situations that help maintain the subtle balance between private and public domains (Schoeman 1984; Schwartz 1968). Spatial privacy, especially privacy in the home, often involves a desire for minimal intrusions from neighbors (e.g., noise, unwanted social contact, nosiness, trespassing) to be able create an island of solitude (Finighan 1979). Children are given daily lessons in the ABC of privacy that demand strict compliance to spatial boundaries (e.g., this is your bedroom, this room is off-limits for you) and teaches them privacy rituals like scheduled bedtime routines (Schwartz 1968). Growing up with the pervasive logic of walls, doorknobs, space dividers, and the hegemony of daily schedules creates deep-seated psychological predispositions as to what constitutes "*my* space" or "*my* time" (Brown et al. 2011; Schwartz 1968; Simmel 1909). With these rules for interpersonal conduct directly built into floorplans and architectural designs, hidden scripts for privacy have become so pervasive that they seem "natural" to the individual (Lindesmith et al. 1999). In this sense, tiny housers are simply Americans who have come to appreciate and expect certain forms of privacy. It is a privacy that respects the individual's personal autonomy (Mangold and Zschau 2019; Newell 1992) and allows them to negotiate access to community on *their terms*.

### 5.3. Closing Thoughts

Lifestyle movements like the tiny house movement illustrate that individual self-interest and the common good may not be mutually exclusive. By expressing a deep desire to live in non-conventional communities, many of these tiny housers articulate a countercultural vision of a more just, more

---

[14]　The Mehinacu Indians of Brazil, for example, live in villages with housing structures arranged in a circular manner in order to foster cohesion and interaction among tribal members. Members retain little privacy especially within their living quarters with upwards of 15 people living in one room (with no kinds of room dividers). Village pathways are arranged so that members can keep a line of sight of one another across the village. The sole privacy granted to embers resides in leaving the village for an extended period to far off secondary living structures. See Altman (1909) for more information.

[15]　Views on privacy can also change over time as the cultural appropriation of Western spatial privacy in contemporary homes in Hong Kong illustrates (see Cheung and Ma 2005).

inclusive, and more collaborative 21<sup>st</sup> century community. Like Thoreau (Thoreau et al. 1906), tiny housers lament the "depersonalization" of contemporary social life and seek a future in which privacy and community can coexist (Moller 1980). Further, probably most importantly, they invite us, like their 18<sup>th</sup> century spiritual mentor, to find our humanity in radically new forms of community.

> "Go not so far out of your way for a truer life—keep strictly onward in that path alone which your genius points out. Do the things which lie nearest to you but which are difficult to do. Live a purer, a more meaningful … life, more true to your friends and neighbors, more noble and magnanimous … To live in relations of truth and sincerity with men is to dwell in frontier country. What a wild and unfrequented wilderness that would be".
>
> (Thoreau et al. 1906)

## 6. Conclusions

Studying tiny housers' views of community satisfies more than mere academic curiosity. It can offer important insights for developers, builders, and policymakers on how to design or redesign the "right" communities. While this study offers a first glimpse into the different communities that tiny housers want to live in, many of the results and conclusions remain tentative. To provide a more comprehensive picture of the community aspects of the TH lifestyle, future research should answer questions such as: How important is living in community for tiny housers in the larger movement? Are tiny housers, as this research seems to suggest, really interested in more cohesive and collaborative communities? If so, what are the primary drivers that shape and maintain these views (e.g., larger social and cultural forces, direct life experiences, boundary flows)? Do tiny housers outside of the United States subscribe to similar notions of community (why, or why not)? Are tiny housers who express no interest in place-based community rugged individualists, or do they merely find a "sense of belonging" in other spaces? How important are virtual post-place communities (e.g., online forums), and what specific roles do they play in tiny housers' lives? Moreover, what are the social and spatial characteristics of existing tiny house communities? Do they mirror the communities described by people in this study, or do their views shift in fundamental ways? Finally, what helps explain the strong desire for privacy even among those who want to live in more collaborative communities? Is it simply an outcome of the spatial realties of tiny living, or should it, as suggested here, be seen as a cultural phenomenon (and thus may vary cross-culturally)? Whatever the nature of future research, it is clear that many intriguing questions remain unanswered.

Mainstreaming non-conventional communities that honor privacy and create true community are difficult to establish. With global capitalism having permeated most organizational and institutional decision-making processes, opposition is often written into the very zoning laws, land use regulations, or lending criteria that so many of us accept as the status quo. Urban and suburban neighborhoods increasingly resemble busy train stations with residents remaining only long enough to follow new transnational credit flows to jobs elsewhere. Without systemic changes and/or large-scale cultural buy-in, TH communities and their respective movements are likely to remain a fringe phenomenon. Efforts by movement activists to change city and state housing ordinances (e.g., Clarkston, Georgia), get states adopt the new IRC tiny house appendix Q (e.g., Massachusetts, California, Idaho, Maine, Oregon, and Georgia), and/or hold large-scale educational events such as tiny house festivals, while commendable, are just not far-reaching enough. Only by putting continued pressure on the legal system, challenging the deeply entrenched cultural meanings of "home," and helping to democratize the logic of housing developments (e.g., via the use of land trusts or other ways of reclaiming the commons) will the tiny house movement be able to remove some of the more intractable obstacles. Good ideas exist and are as diverse as inclusive intergenerational cohousing models, mixed-used green housing developments (advocated by visionaries of the New Urbanism School), or radical communes built around regenerative designs (as promoted by some activists within the de-growth or

alter-globalization movements). While the future remains uncertain, solving humanity's challenges will require for us to fundamentally re-imagine what it means to live together.

**Author Contributions:** All three authors have contributed equally to the research and manuscript (i.e., conceptualization, methodology, formal analysis, investigation, resources, data curation, writing—original draft preparation, writing—review and editing, visualization, project administration, and securing funding). All authors have read and agreed to the published version of the manuscript.

**Funding:** This research was partially funded through the 2017 UNG Summer FUSE Program and generous support from the Department of Sociology and Human Services of the University of North Georgia.

**Acknowledgments:** The authors would like to extend our gratitude to the participants for their time and willingness to share their views on community with us. We are also deeply indebted to Codey Collins, Dace Lewis, Devin Hing and Valerie Odorico for their crucial role in the interview and/or transcription process. We are appreciative of Bryan Dawson's and Abby Meyer's assistance in running and interpreting the ANOVAs, as well as John Dewey for his help with choosing adequate statistical techniques for non-random samples. In addition, we would like to thank the four anonymous reviewers for providing extremely helpful and constructive feedback on earlier versions of the manuscripts. Finally, we are grateful to Barb Wilson and Sue Mattison for proofreading the manuscript. The Institutional Review Board (IRB) of the University of North Georgia approved this study under code 2017-118-COM.

**Conflicts of Interest:** The authors declare no conflict of interest.

## Appendix A

**Table A1.** The Social Profile of Participants.

| Name | Age | Gender | Income | Education | State | Location [1] | Politics [2] | Religion [3] |
|---|---|---|---|---|---|---|---|---|
| Abraham | 55+ | Male | $34,000–$49,999 | HS Diploma | VA | Rural | Other | Christian |
| Archie | 35–54 | Male | Less than $25,000 | Master's Degree | TX | Rural | Liberal | Buddhist |
| Artemis | 18–34 | Female | $50,000–$74,999 | College Degree | MN | Suburban | Conservative | Christian |
| Ashley | 18–34 | Female | $25,000–$34,999 | College Degree | TX | Suburban | Conservative | Christian |
| Barbara | 18–34 | Female | $25,000–$34,999 | College Degree | n/a * | Rural | Other | Atheist |
| Ben | 35–54 | Male | $50,000–$74,999 | Master's Degree | SC | Suburban | Liberal | Agnostic |
| Canan | 55+ | Male | $75,000–$99,999 | Some College | VA | Rural | Liberal | Other |
| Dan | 35–54 | Male | $35,000–$49,999 | College Degree | GA | Suburban | Conservative | Other |
| Frank | 18–34 | Male | Less than $25,000 | College Degree | n/a * | Urban | Liberal | Agnostic |
| Greta | 35–54 | Female | $50,000–$74,999 | College Degree | NY | Urban | Liberal | Agnostic |
| Jane | 55+ | Female | $25,000–$34,999 | College Degree | n/a * | Rural | Other | Spiritual |
| Jenny | 35–54 | Female | $35,000–$49,999 | Certifications [4] | TX | Suburban | Libertarian | Christian |
| John | 55+ | Male | $25,000–$34,999 | PhD Degree | n/a * | Urban | Other | Spiritual |
| Mary | 55+ | Female | $75,000–$99,999 | College Degree | n/a * | Rural | Liberal | Christian |
| Namor | 18–34 | Male | $50,000–$74,999 | Master's degree | AL | Suburban | Other | Christian |
| Nancy | 35–54 | Female | $50,000–$74,999 | College Degree | MN | Rural | Other | Christian |
| Rick | 18–34 | Male | $75,000–$99,999 | College Degree | GA | Rural | Conservative | Christian |
| Roselle | 18–34 | Female | $25,000–$34,999 | Certifications [4] | OR | Rural | Liberal | Other |
| Samantha | 55+ | Female | $34,000–$49,999 | Some College | n/a * | Rural | Other | Christian |
| Sebastian | 35–54 | Male | $25,000–$34,999 | Master's Degree | CO | Other | Liberal | Other |
| Tessa | 18–34 | Female | $25,000–$34,999 | College Degree | TX | Urban | Other | Agnostic |
| Tim | 55+ | Male | $50,000–$74,999 | MBA Degree | CA | Suburban | Liberal | Spiritual |
| Tom | 35–54 | Male | $75,000–$99,999 | Master's Degree | SC | Urban | Other | Spiritual |
| Venus | 55+ | Female | Less than $25,000 | PhD Degree | TN | Rural | Other | Buddhist |

**Note:** 1 ... current home location, 2 ... political self–identification, 3 ... religious orientation, 4 ... trade school, technical school, or vocational training, n/a* information not available (some declined to answer this question).

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
