# Peer review of "Small Houses, Big Community: Tiny Housers’ Desire for More Cohesive and Collaborative Communities"

_socsci, doi:10.3390/socsci9020016_

Round 1
Reviewer 1 Report
The article analyses the movement of Tiny Houses and explores how the importance of the sense of community for people living in this kinds of accommodations. It argues that despite the fact that these residents are choosing to live in individual homes, they are nevertheless driven by an aspiration to a dynamic social and community life. The paper also discusses the tension between their request for privacy and this community aspiration. The results are discussed in relation to different kinds of theoretical perspectives on the evolution of communities, including pessimistic accounts on the decline social capital, or on the opposite, works that insist how place can help shape new forms of communities.
The paper is original, it discusses relevant literatures, and has a clearly detailed methodological approach. Nevertheless, there are some issues that need to be addressed.
First of all, the research design raises some questions.
-There is a selection bias: the informants interviewed having been recruited on social media groups, they are more likely to value sense of community, than if other sampling methods had been used. This is a problem especially for quantitative analyses, because these interviewees cannot be considered to be representative. To the least, this issue needs to be acknowledged.
-An expected method for this type of research question would have been to conduct observations. The paper provides no account of any observation, or description of Tiny Housing estates. This relate to another point below.
-The answers may vary widely based on the location of the Tiny house dweller. Is there data on location and how this may have affected answers? Another important aspect which is absent is the social profile of the respondents. According to the literature, this may greatly affect the kinds of community aspirations and behaviours, so this needs to be documented in some way.
- When reading the responses of the informants, they seem to be talking about what they would expect in terms of community in Tiny house neighbourhoods, instead of about the actual community activities or relations that they are developing. Their accounts would be much more relevant if they refer to tangible actions rather than opinions. Also, a key information we would need to have is their motivations in settling in this kind of accommodation, and to what extent the sense of community has been an argument for them.
- Final point on the research design, it is not clear whether there is really specificity in the responses of Tiny House dwellers as opposed to people living in regular houses or apartments. Of course, having a control sample may be too heavy, but at least, the answers collected should be discussed in relation to similar works conducted on people living in different kinds of housing.
Second, the reader lacks of background on the Tiny House movement.
-We need figures: How many Tiny houses have been built?
-We need location patterns: where are Tiny Houses located? To the least, we would need to know the cities of the informants that have been interviewed, what kinds of urban environment the Tiny House are located in.
-We need also information on governance and planning logics: one of the informants mentions a real-estate developer: we need background on who develop these projects, who own the land, how these areas are managed?
-Also a key question is the reasons why people choose to live in such types of accommodation. And what I am wondering is especially if there is not an economic factor which could be prevalent. This is a major question to answer before even asking what their sense of community is.
And third, some theoretical clarifications would be needed.
-3 types of literatures on communities are identified. This is initially presented as a debate between three perspectives. So one expects the paper to provide support in favour of one of the perspectives over the other. But the paper does not clearly come back to this literature to state how it has contributed to the debate and provided evidence to support one perspective over the other.
-In the literature review, there are also several developments on various forms of alternative and community urbanism like cohousing, ecovillage. But it is not very clear how this connects to the Tiny House movement. Again, this is also because we lack of background information on the Tiny house movement and the motivations of people living in these types of accommodations.
-The authors seem to suggest at various occurrences that the Tiny House movement is inspired by Henry David Thoreau: this is an argument made at several times. On what is it based? This seems quite surprising as they are not presented as separate from the city. The paper seems also to suggest that living in Tiny Houses is unconventional, but does not provide much elements to back this argument. The reference to Thoreau seems to imply a rather radical opposition to social norms, is it the case?
Reviewer 2 Report
Rather than citing a literature review, the essay would do well to engage and cite the actual primary sources (thus avoiding repeating the gaps in research and interpretation from the secondary source, ie. the work on Thoreau and the tiny house movement - robustly analyzed in Anson's later publications - or the international iterations of the movement and the relationship to global capital, which has much research available) In general, the research should be deepened. The survey sample of 24 people feels rather small for the size and diversity of the movement itself. The phrase "Tiny housers, like their American counterpart" seems to suggest that tiny housers do not exist in America.
Author Response
Response to Reviewer Two Comments
Point 1: “Rather than citing a literature review, the essay would do well to engage and cite the actual primary sources (thus avoiding repeating the gaps in research and interpretation from the secondary source, ie. the work on Thoreau and the tiny house movement - robustly analyzed in Anson's later publications - or the international iterations of the movement and the relationship to global capital, which has much research available)”
Response: We have taken a close look at the work of April Anson as suggested by the reviewer and have found her recent work in line with many of our own arguments. We have therefore tried to weave some of her arguments into both the introduction as well as the discussion of the manuscript. We have also inserted a direct quote from Thoreau to further bolster the argument on his ‘unconventional’ form of community and some of Anson’s 2017 work The Patron Saint of Tiny Homes. We have also tried to touch on the idea of global capital and connections to the global degrowth movement. However, it is our understanding of the literature that there are no formal or even stable unformal links that exist between the global degrowth movement and the tiny house movement (a point picked up by Anson’s 2018 piece entitled Framing degrowth: The radical potential of tiny house in which she makes a convincing argument that both movements have enough teleological affinities that they can learn from one another). So while we do appreciate the reviewer’s insights into this issue, we do not think that further exploration of global capital dynamics and the degrowth movement would add much substance to our actual arguments (which primarily focus on views of community). We have, however, added additional references to international iterations of the tiny house movement in the introduction and provided examples of tiny house movements in countries like Germany, Australia, or the Netherlands. We hope that these changes have addressed most of the reviewer's concerns.
Point 2: “the research should be deepened”
Response: We appreciate the feedback and we have made a concerted effort to deepen the analysis and interpretation of the data by drawing on a wider range of theoretical and empirical studies. In deepening the research and better contextualizing the findings for the reader, we have provided more background information on the formation of the tiny house movement. We have also strengthened connections to existing lifestyle and communal movements throughout the discussion and conclusion. While we made efforts to deepen the analysis theoretically, collecting additional data is simply beyond scope of the project at this time. We hope the reviewer appreciates our dilemma in this case.
Point 3: “The survey sample of 24 people feels rather small for the size and diversity of the movement itself”
Response: We agree with the reviewer that the sample is small. However, we would like to point out that our sampling design follows best practices within qualitative and mixed methods research and was geared toward improving sample representativeness. We have added footnote 8 to provide a more detailed rationale for the sample size and our claims to improved representativeness (over simple convenience sampling techniques). While the reviewer is correct that the movement is very complex, we would like to add that our sample only captures a segment of the overall transnational movement (US tiny housers with interest in community). While other research (e.g. (Boeckermann 2017)) and some of our own unpublished work of a US-based survey involving 453 participants (manuscript in preparation) substantiates the findings in this study, we have opted to stay clear of strong claims to external validity throughout the paper. We have made, wherever appropriate, the language of our conclusions more tentative and pointed out that the research is exploratory in nature.
Point 4: “The phrase "Tiny housers, like their American counterpart" seems to suggest that tiny housers do not exist in America”
Response: We agree that the phrase could potentially be confusing. The statement has been changed to read “tiny housers, like many of their American contemporaries …”.
References
Anson, April. 2018. "Framing Degrowth: The Radical Potential of Tiny House Mobility." Pp. 68-79 in Housing for Degrowth: Routledge.
Boeckermann, Lauren Michelle. 2017. "Dreaming Big and Living Small: Examining Motivations and Satisfaction in Tiny House Living " Senior, University of South Carolina.

Reviewer 3 Report
Besides saying how TH or other trends gained popularly 'recently', can explain more on the historical context and give evidence from what happens in society
No emphasis on discussing how conclusion is based on results of statistical significance - why?
Seems like the results are pretty optimistic given it's a small sample research.
Non-random sampling - is it ok to use these statistical analyses?
Reviewer 4 Report
The paper entitled "Small Houses, Big Community: Tiny Housers’ Desire for More Cohesive and Collaborative Communities" focuses (as it quite long title suggests) on testing the hypothesis whether tiny housers do entertain an interest in community.
The "tiny house" movement that the author(s) claim "has gained traction across the world over the past two decades" is truly an interesting phenomenon. However, it should be properly discussed whether it is targeted at living a sustainable life, or just living in a trailer park due to the lack of money and perspectives. It is notable that the TH movement is especially popular in the U.S. Can the author(s) elaborate on that a little bit more?
The paper is interesting but it needs some serious improvements before it can be published:
1) The Literature review needs to be made as a separate section. It should be linked to the overall discussion and introduction.
2) The methodology and the data collection need to be clarified. The paper claims to use the data from the interviews with the "twenty-four community-oriented tiny housers". Who were these people? Which country they are from? How was the selection arranged (a snowball, perhaps)? All of these need to be made clear.
3) Are the results country-specific? People in Europe have tiny houses as their holiday homes. People in Asia and Africa all have tiny houses because they are poor and cannot afford anything else. Is there any difference between this and the U.S.? Is TH phenomenon a product of luxury life or a by-product of despair?
4) Citations from the interview might need to be moved into the appendix. They are numerous and make it hard to follow what the main paper is saying.
5) Conclusions need to be extended and policy implications need to be drawn.
6) The paper should be checked by the native English speaker. There are some minor issues such as: "tiny housers were interview about their views" (instead of "interviewed"), etc.
Round 2
Reviewer 1 Report
The additional background is much appreciated. It makes the article interesting for a wider audience. My main questions on the methodology have been addressed. More minor issues have been left for future research. The theoretical discussion is clearer, especially the relation to Thoreau's thought.
Reviewer 3 Report
A gentle highlight to the main results would help.
Reviewer 4 Report
The paper has been sufficiently modified and all comments have been thoroughly addressed. The Introduction has been extended and methodology has been explained. The Literature review was made into a separate section and other issues were also tackled. I think that the paper can be accepted in its current form.
